# O-GlcNAc signaling increases neuron regeneration through one-carbon metabolism in *Caenorhabditis elegans*

**Dilip Kumar Yadav[1]\*, Andrew C Chang[1], Noa WF Grooms[2], Samuel H Chung[2], Christopher V Gabel[1,3]\***

[1]Department of Pharmacology, Physiology and Biophysics, Chobanian & Avedisian School of Medicine, Boston University, Boston, United States; [2]Department of Bioengineering, Northeastern University, Boston, United States; [3]Neurophotonics Center, Boston University, Boston, United States

**Abstract** Cellular metabolism plays an essential role in the regrowth and regeneration of a neuron following physical injury. Yet, our knowledge of the specific metabolic pathways that are beneficial to neuron regeneration remains sparse. Previously, we have shown that modulation of O-linked β-N-acetylglucosamine (O-GlcNAc) signaling, a ubiquitous post-translational modification that acts as a cellular nutrient sensor, can significantly enhance in vivo neuron regeneration. Here, we define the specific metabolic pathway by which O-GlcNAc transferase (*ogt-1*) loss of function mediates increased regenerative outgrowth. Performing in vivo laser axotomy and measuring subsequent regeneration of individual neurons in *C. elegans*, we find that glycolysis, serine synthesis pathway (SSP), one-carbon metabolism (OCM), and the downstream transsulfuration metabolic pathway (TSP) are all essential in this process. The regenerative effects of *ogt-1* mutation are abrogated by genetic and/or pharmacological disruption of OCM and the SSP linking OCM to glycolysis. Testing downstream branches of this pathway, we find that enhanced regeneration is dependent only on the vitamin B12 independent shunt pathway. These results are further supported by RNA sequencing that reveals dramatic transcriptional changes by the *ogt-1* mutation, in the genes involved in glycolysis, OCM, TSP, and ATP metabolism. Strikingly, the beneficial effects of the *ogt-1* mutation can be recapitulated by simple metabolic supplementation of the OCM metabolite methionine in wild-type animals. Taken together, these data unearth the metabolic pathways involved in the increased regenerative capacity of a damaged neuron in *ogt-1* animals and highlight the therapeutic possibilities of OCM and its related pathways in the treatment of neuronal injury.

**\*For correspondence:**
dyadav1@bu.edu (DKY);
cvgabel@bu.edu (CVG)

**Competing interest:** The authors declare that no competing interests exist.

## Editor's evaluation

This important work reveals that increased flux towards one carbon metabolism improves neuronal regeneration after injury in *C. elegans*. The presented data are solid and provide compelling support for this conclusion.

## Introduction

To regenerate efficiently, a damaged neuron must undergo molecular and metabolic rearrangement to induce and endure a range of complex cellular processes (*Mahar and Cavalli, 2018*; *Taub et al., 2018*; *Yang et al., 2020*). These processes are extremely metabolically challenging, energy demanding, and critical for the regenerative capacity of a neuron (*Byrne et al., 2014*; *Cartoni et al., 2016*; *He and Jin, 2016*; *Yang et al., 2020*). The importance of metabolic pathways, particularly in

neuronal regeneration including the insulin-signaling pathway, energy metabolism, and mitochondrial function have been reported in research articles by several groups (*Byrne et al., 2014*; *Cartoni et al., 2016*; *Han et al., 2016*; *Han et al., 2020*). Nonetheless, critical questions remain as to the alterations in cellular metabolism and metabolic pathways linked with energy production in a damaged and regenerating neurons and how these processes might be exploited for therapeutic benefits.

In a previous study our group demonstrated that perturbation in O-GlcNAc signaling, a post-translational modification of serine and threonine that is known to act as a nutrient sensor, substantially increased axonal regeneration in *Caenorhabditis elegans* (*C. elegans*) (*Taub et al., 2018*). Carrying out in vivo laser axotomies, we demonstrated that a reduction of O-GlcNAc levels, due to deletion mutation of the *ogt-1*, induces the AKT-1 branch of the insulin-signaling pathway to utilize glycolysis and significantly enhanced neuronal regeneration. Inhibition of the glycolytic pathway through RNAi knockdown of phosphoglycerate kinase (*pgk-1*) or loss of function of phosphofructokinase-1.1 (*pfk-1.1*) specifically suppressed *ogt-1* enhanced regeneration but did not alter wild-type regeneration (*Taub et al., 2018*). Furthermore, supplementation with glucose in wild-type animals is sufficient to increase axonal regeneration after axotomy (*Taub et al., 2018*). These observations established the significance of glycolytic metabolism to control and enhance neuronal regeneration.

To date key questions remain as to what specific metabolic pathways are stimulated in the *ogt-1* mutant background and what cellular processes are augmented to increase regenerative capacity. Numerous reports suggest that increased glycolysis averts metabolic flux toward OCMto regulate numerous biological processes including molecular reprogramming, immunological functions as well as neuronal development and function (*Iskandar et al., 2010*; *Konno et al., 2017*; *Yu et al., 2019b*). In addition, studies have reported the importance of metabolic amendments of OCM, the SSP and the TSP in neuronal development, structure, function, and regeneration (*Iskandar et al., 2010*; *Bonvento and Bolaños, 2021*; *Lam et al., 2021*; *Chen et al., 2022*). Measuring neuronal regeneration in *C. elegans* following laser axotomy under genetic and pharmacological perturbations of metabolic pathways, we demonstrate that both functional OCM and glycolytic flux towards OCM *via* the SSP are essential for enhanced regeneration in *ogt-1* animals. From there, we observed that metabolic pathways from OCM through the TSP, that result in cystathionine metabolism into Acetyl-CoA *via* the vitamin B12 independent shunt pathway, is also critical. Taken together our results illustrate how *ogt-1* acts as a major regulator of metabolic pathways to orchestrate and maximize the regenerative response in a damaged neuron and suggest that OCM and its related pathways could serve as a potent neurotherapeutic target.

## Results

### Blocking the hexosamine biosynthesis pathway (HBP) is adequate to phenocopy the enhanced neuronal regeneration of *ogt-1* animals

Following our previous study, *Taub et al., 2018*, we sought to confirm and expand on our finding that *ogt-1* loss of function mutation enhances neuron regeneration through modulation of glycolysis. Performing laser axotomy on individual neurons in vivo and measuring regenerative outgrowth after 24 hr, we found that either *ogt-1* (deletion, ok1474 strain crossed with zdis-5) mutation or the enzymatically *ogt-1* dead allele (ogt-1-dAl; OG1135 strain crossed with zdis-5) equally increase neuronal regeneration following laser axotomy of mechanosensory neurons in *C. elegans* (*Figure 1A–B*, *Figure 1—source data 1*). Reduced O-GlcNAc levels due to the loss of function of *ogt-1* will effectively block the metabolic flux into the HBP, diverting metabolites towards glycolysis (*Yi et al., 2012*; *Jóźwiak and Forma, 2014*; *Kim et al., 2018*). To investigate if blocking the HBP is sufficient to enhanced neuronal regeneration in the wild-type worms, we knocked down Glutamine-Fructose 6-phosphate Amino Transferase (*gfat-1* and *gfat-2*) using neuron-specific RNAi. *gfat-1* and *gfat-2*, orthologs of the human glutamine-fructose-6-phosphate transaminase 1 (*GFPT1*), catalyzes the very first and rate limiting step of HBP (*Figure 1A*). We found that knocking down either *gfat-1* or *gfat-2* in the wild type worms, significantly increase the regeneration of mechanosensory neurons, similar to *ogt-1* loss of function, (*Figure 1C*, *Figure 1—source data 1*).

Earlier we reported that the regenerative effects of *ogt-1* were dependent on numerous elements of the glycolytic pathway. In this study, we tested pyruvate kinase (PK), PKM1/2 (encoded by *pyk-1* in *C. elegans*) which catalyzes the final step of glycolysis and is known to be regulated by O-GlcNAc

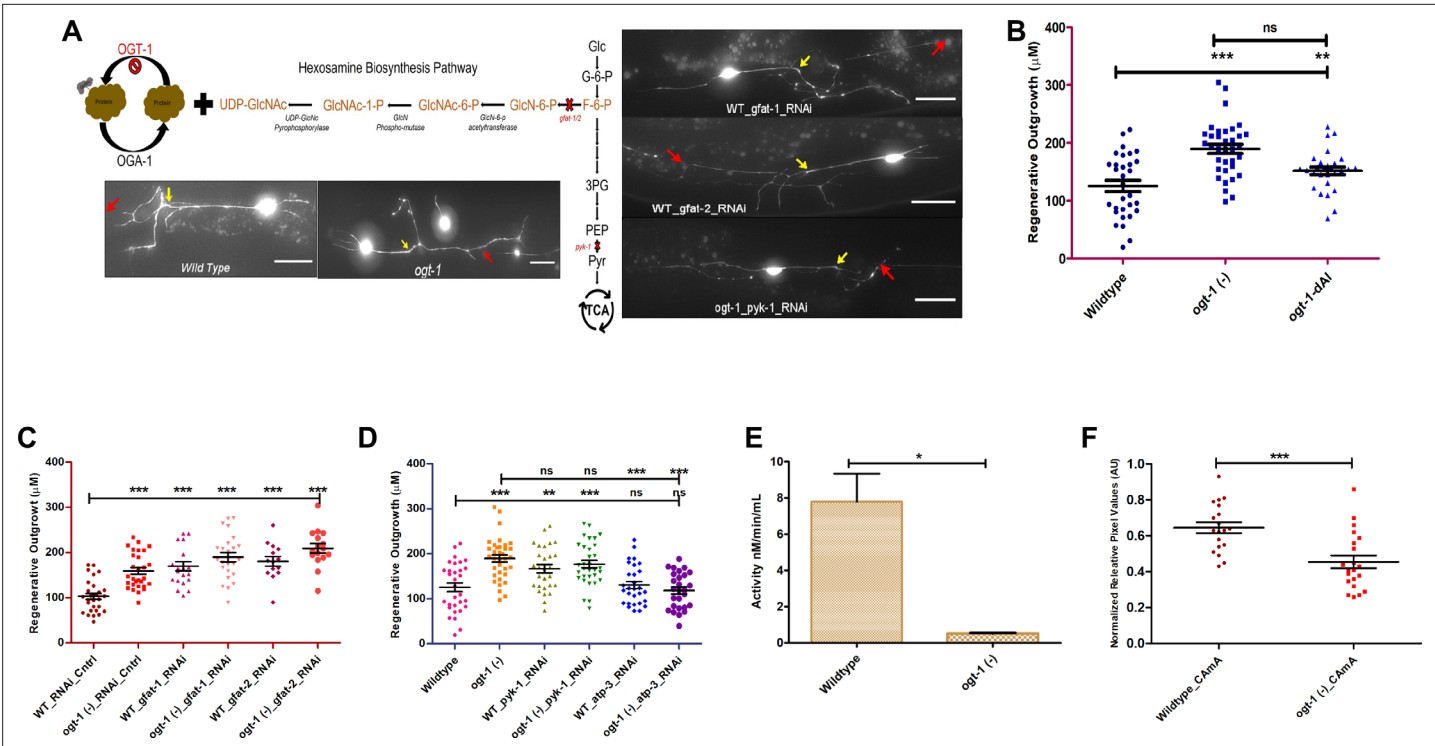

**Figure 1.** Blocking the HBS pathway is sufficient to phenocopy the neuronal regeneration of O-GlcNAc transferase (*ogt-1)* animals. (**A**) Schematic diagram showing the hexosamine synthesis pathway linking glycolysis and *ogt-1* function, and the representative image of the effect of *ogt-1* mutation and *gfat-1/gfat-2* and *pyk-1* RNAi knockdown on regenerating neurons imaged at 24 hr (yellow arrow indicates the proximal and red arrow indicated distal point of injury). (**B**) 24 hr regeneration data of wild-type (WT), *ogt-1* deletion mutant (OGT-1) and *ogt-1* dead allele (OGT-1-dAl, strain OG1135) worms on nematode growth media (NGM) after 24 hr. (**C**) 24 h regeneration data of control and *gfat-1/gfat-2* RNAi experiments. (**D**) 24 hr regeneration data of control and RNAi experiment for *pyk-1* and *atp-3*. (**E**) pyk-1 activity measured in WT and *ogt-1* animal whole lysate using pyruvate kinase (PK) Assay Kit (Abcam, cat# Ab83432). (**F**) Relative amount of ATP measured using a fluorescence resonance energy transfer (FRET)-based ATP sensor. OGT-1-dAl; ogt-1-dead Allele, AU; Arbitrary Unit, scale bar = ~10 μM, all data shown in ± SEM, analytical methods student t-test and one-way ANOVA, *p <0.05, **p <0.01, ***p <0.001.

The online version of this article includes the following source data and figure supplement(s) for figure 1:

**Source data 1.** Regeneration lengths measured with ImageJ/FIJI and ATP levels measured using ATP sensor data for *Figure 1*.

**Figure supplement 1.** CeNGEN pyk expression pattern in the neurons of *C. elegans*, ATP levels and ATP utilization in Wildtype and *ogt-1* animals.

levels (*Wang et al., 2017*; *Bacigalupa et al., 2018*; *Yu et al., 2019a*). *C. elegans* has two orthologs of mammalian PK, *pyk-1* and *pyk-2*, with *pyk-1* expression primarily in neurons including the mechano-sensory neurons ALM and PLM and *pyk-2* showing limited neuronal expression (*Hammarlund et al., 2018*; *Figure 1—figure supplement 1A* and *Figure 1—figure supplement 1B*). Interestingly, we found that the knock down of *pyk-1, via* neuron-specific RNAi, does not affect the enhanced regeneration in the *ogt-1* mutant but significantly increases regeneration in WT (*Figure 1A and D*, *Figure 1—source data 1*), to levels similar to that of *ogt-1*. Furthermore, performing *pyk-1* activity assay in whole worm lysate we observed that over all *pyk-1* activity is significantly down in *ogt-1* worms (*Figure 1E*). These results suggest the final step of glycolysis, mediated by PYK-1, is not in fact involved in the enhanced regeneration of the *ogt-1* animals.

To investigate if energy production is critical for enhanced regeneration in *ogt-1* mutant animals, we performed neuron-specific RNAi knockdown of *atp-3*, an ortholog of human ATP5PO (ATP synthase peripheral stalk subunit OSCP) predicted to have proton-transporting ATP synthase activity. *atp-3* knockdown abrogated the *ogt-1*-mediated enhanced regeneration but had no effect on regeneration in WT (*Figure 1D*, *Figure 1—source data 1*). This is consistent with a critical role of ATP production in the enhanced regeneration of *ogt-1*. However, these effects did not translate to whole animal ATP level measurements. Employing a FRET-based transgenic fluorescence ATP sensor (as described earlier in *Soto et al., 2020*; *Figure 1F*, *Figure 1—source data 1*) as well as ATP measurements

in whole worm lysate, we found that ATP levels were significantly lower in *ogt-1* than WT worms (*Figure 1—figure supplement 1C*). In addition, we assessed the ATP utilization from whole animals by measuring the pyrophosphate (PPi) levels as an indirect indication of ATP utilization but found no measurable difference between WT and *ogt-1* worms (*Figure 1—figure supplement 1D*). Taken as a whole, these results infer that metabolic flux modulation through the majority of the glycolytic pathway and neuron-specific ATP production is important for *ogt-1*-mediated enhanced regeneration, but a complex interaction of metabolic pathways beyond that of canonical glycolysis may be involved specifically within the damaged and regenerating neurons.

## Gene expression analysis reveals the involvement of OCM and its offshoot pathways in enhanced neuron regeneration in *ogt-1* animals

To identify additional genes and pathways involved in the enhanced regeneration of the *ogt-1* background, we took an unbiased approach measuring differential gene expression *via* RNA-seq analysis in WT and *ogt-1* mutants. We first executed RNA-seq analysis from RNA isolated from whole animals and identified a substantial number of differentially expressed genes (DEGs) in *ogt-1* compared to WT (*Figure 2A*, *Figure 2—source data 1*). Gene ontology (GO) and KEGG pathway classification analysis of DEGs identified metabolic processes such as carbohydrates, lipids, amino acids, and nucleotide metabolism as the most enriched biological processes (*Figure 2B–C*, *Figure 2—source data 1*). In addition, cell membrane, cargo transport, nutrient reservoir, and energy metabolism are also enriched in *ogt-1* (*Figure 2B–C*, *Figure 2—source data 1*). KEGG metabolic pathway enrichment analysis revealed the enrichment of xenobiotics, drug metabolism along with glutathione metabolism, energy metabolism, amino acid, and nitrogen metabolic pathways (*Figure 2D*, *Figure 2—source data 1*). GO molecular function analysis highlights the nutrient reservoir, glutathione, and s-adenosyl methionine (SAM) dependent molecular functions (*Figure 2E*, *Figure 2—source data 1*). The enrichment of amino acid, nucleotide, glutathione, and SAM-dependent metabolic pathways indicate a possible role of OCM and its offshoot pathways in *ogt-1* mutant mediated regeneration.

To further investigate if OCM and its related pathways are influenced by *ogt-1* mutation specifically within neuronal cells, we performed RNA-seq analysis in the RNA samples isolated from FACs (Fluorescence-activated cell sorting) sorted neuronal cells in WT and *ogt-1* worms (*Figure 2—figure supplement 1A*). Neuron-specific RNA-seq analysis identified a significant number of DEGs (*Figure 2—figure supplement 1B*, *Figure 2—source data 2*). Interestingly, we did find that genes linked to HBP and glycolysis such as *gaft-1/gfat-2* and *pyk-1* were differentially regulated in our neuron-specific RNA-seq data (*Figure 2—source data 2*). As with whole worm analysis, GO pathway analysis of neuron-specific DEGs identified metabolic processes such as cellular, macromolecule, nitrogen compound, nucleic acid metabolism, *etc.* (*Figure 2F*, *Figure 2—source data 2*). GO analysis of twofold up-regulated genes revealed neuron-specific pathways as anticipated (neuronal perception, chemical and olfactory perception, synapses, *etc.*) along with carbohydrate and polysaccharide metabolic pathways (*Figure 2—figure supplement 1C*, *Figure 2—source data 2*), while twofold down-regulated genes included biological processes like meiosis, mitosis, gamete/germ cell production and maturation, reproduction, cell cycle, nuclear division, and embryonic developments, *etc.* which are expected to be down regulated in the neuronal tissue (*Figure 2—figure supplement 1D*, *Figure 2—source data 2*). Our top 50 up and down-regulated genes (*Figure 2—figure supplement 1E* and *Figure 2—figure supplement 1F*) include important genes regulated by *daf-2* and *daf-16* which have been reported to play a critical role in adult neuron function and regeneration (*Kaletsky et al., 2016*). In addition, other important genes involve in metabolism, epigenetic modification, and ATP metabolism are also enriched. Employing whole animal qRT-PCR, we further confirmed that *folr-1, metr-1, sams-1,* important genes for OCM, were significantly up-regulated in the *ogt-1* background compared to WT (*Figure 2G*). While DNA methyltransferase (*damt-1*) was significantly down regulated and DNA demethylases (*nmad-1*) was unchanged (*Figure 2G*), Further bioinformatic analysis of neuron-specific DEGs using the Functional Annotation Tool 'DAVID Bioinformatics Resources' revealed the enrichment of metabolic pathways such as glycolysis, lipid metabolism along with SSP, OCM, amino acid, nucleotide, and nitrogen compound metabolism, *etc.* (*Figure 2—figure supplement 2A*). While biosynthesis of cofactor analysis specified enrichment of Folate, Methionine, and SAM metabolism cycles, glutathione metabolism, and ATP synthesis pathways (*Figure 2—figure supplement 2B*). Taken together, the results of our unbiased high throughput gene expression analysis strongly indicate

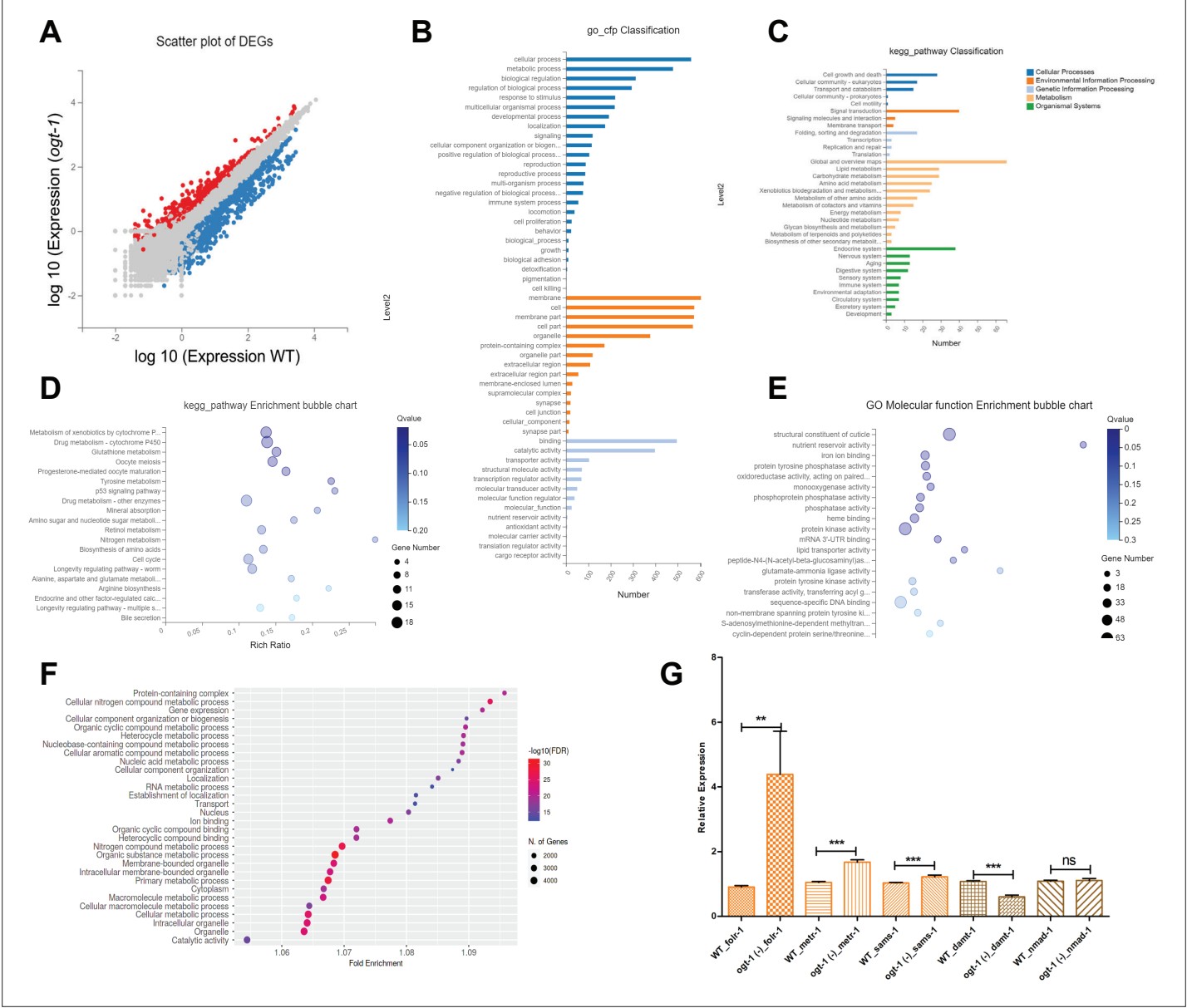

**Figure 2.** RNA sequencing (RNA-seq) data analysis suggests important role of one-carbon metabolism (OCM) and related pathways in O-GlcNAc transferase (*ogt-1*)-mediated neuronal regeneration. (**A**) A scatter plot of differentially expressed genes (DEGs) identified in RNA-seq between wild-type (WT) and *ogt-1* mutants. (**B**) Gene Ontology (GO) classification of DEGs in WT-vs-*ogt-1*. (**C**) KEGG pathway classification of DEGs in WT-vs-*ogt-1*. (**D**) KEGG pathway enrichment bubble plot of DEGs. (**E**) Enrichment bubble plot of GO molecular function analysis DEGs. (**F**) GO analysis of DEGs identified in neuron-specific RNA-seq between WT and *ogt-1* mutant (FDR0.1). (**G**) qRT-PCR of selected genes involved in one-carbon metabolism (OCM) (*folr-1, metr-1, and sams-1*) and nucleic acid methyltransferases and demethylases (*damt-1 and nmad-2*). All data shown ± SEM, Student t-test; *p<0.05, **p<0.01, ***p<0.001.

The online version of this article includes the following source data and figure supplement(s) for figure 2:

**Source data 1.** List of DEGs, GO and KEGG enrichment analysis of from whole body RNAseq data.

**Source data 2.** List of DEGs, genes showing 2 fold expression changes and GO enrichment analysis form neuronal RNAseq Data.

**Figure supplement 1.** FACs shorting, Neuronal RNAseq DEGs, Gene Ontology (GO) pathways analysis and list of top 50 DEGs.

**Figure supplement 2.** Metabolic Pathway Analysis using DAVID from neuronal differentially expressed genes from neuronal RNAseq.

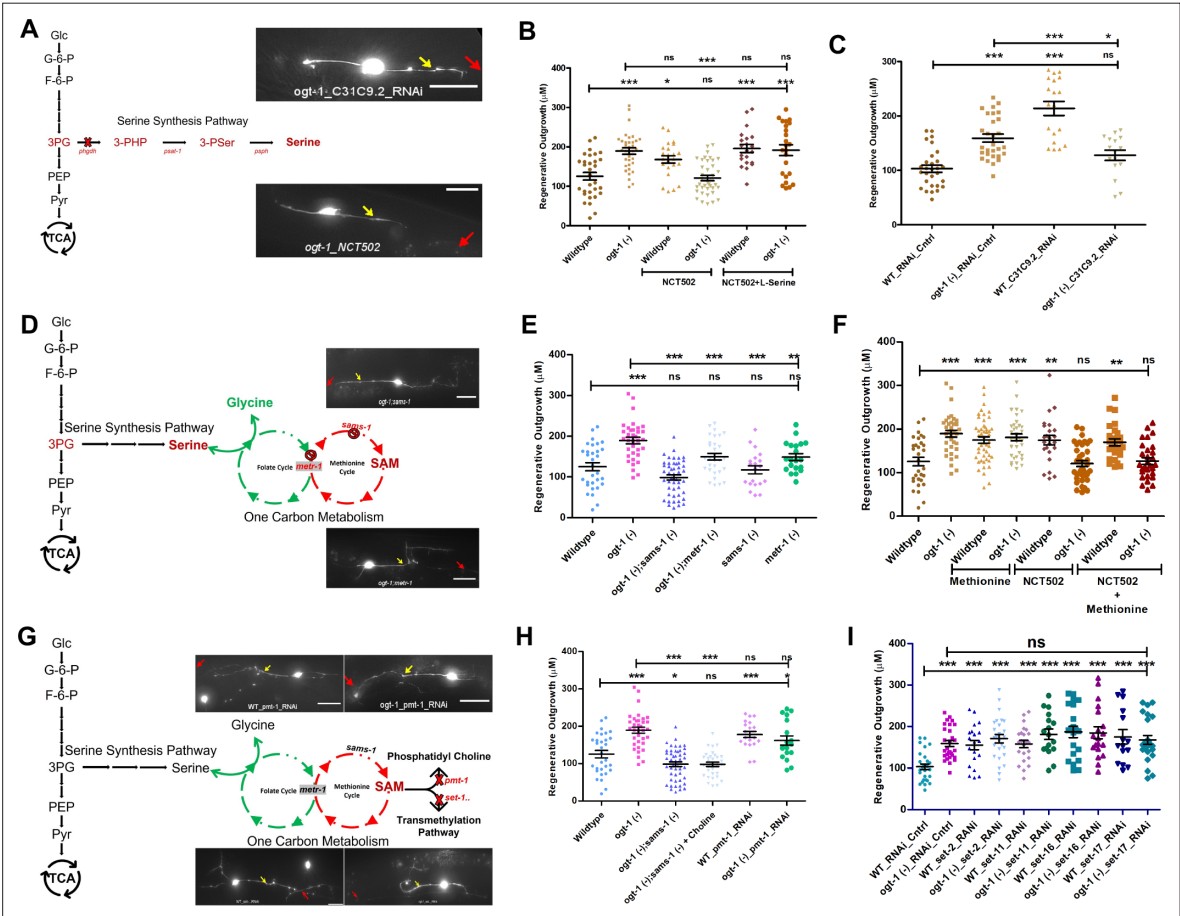

**Figure 3.** Functional one-carbon metabolism (OCM) and serine synthesis pathway (SSP) are essential for neuronal regeneration in O-GlcNAc transferase (*ogt-1)* worms. (**A**) Schematic representation showing glycolysis and the SSP, along with representative images at 24 hr neuron regeneration in conditions blocking the SSP in *ogt-1* mutants using either neuron-specific RNAi or NCT502 drug (yellow arrow indicates the proximal and red arrow indicated distal point of injury). (**B**) Effect of NCT502 drug and supplementation of serine on wild-type (WT) and *ogt-1* mutant 24 hr neuronal regeneration. (**C**) Effect of neuron-specific RNAi against *C31C9.2* (ortholog of human *PHGDH* gene) on WT and *ogt-1* mutant neuronal regeneration. (**D**) Schematic representation of the metabolic link between glycolysis and OCM via SSP, along with representative images of 24 hr neuron regeneration with different OCM gene mutations in *ogt-1* background (yellow arrow indicates the proximal and red arrow indicated distal point of injury). (**E**) Effects of *metr-1* and *sams-1* mutations on enhanced regeneration in *ogt-1* worms. (**F**) Effects of methionine supplementation on regeneration in WT, *ogt-1* animals, and on the *phgdh-1* inhibitor drug NCT502. (**G**) Schematic representation of OCM metabolite SAM usage in lipogenesis and transmethylation, along with representative images of neuron regeneration when they are blocked (yellow arrow indicates the proximal and red arrow indicated distal point of injury). (**H**) 24 hr neuron regeneration with choline supplementation in *ogt-1/sams-1* dual mutant and neuron-specific RNAi against *pmt-1*. (**I**) 24 hr neuron regeneration when blocking methyltransferases by neuron-specific RNAi (*set-2, set-11, set-16,* and *set-17*) in WT and *ogt-1* animals. scale bar = ~10 μM, all data shown in ± SEM, one-way ANOVA *p<0.05, **p<0.01, ***p<0.001.

The online version of this article includes the following source data and figure supplement(s) for figure 3:

**Source data 1.** Regeneration lengths measured with ImageJ/FIJI data for *Figure 3*.

**Figure supplement 1.** Effect of Serine, NCT502 and methionine supplementation on Neuronal regeneration and pyk-1 activity, systemic RNAi of C31C9.2 and Expression patter important OCM genes from Neuronal RNAseq.

**Figure supplement 1—source data 1.** Regeneration lengths measured with ImageJ/FIJI data for *Figure 3—figure supplement 1A*.

**Figure supplement 1—source data 2.** Regeneration lengths measured with ImageJ/FIJI data for *Figure 3—figure supplement 1B*.

**Figure supplement 1—source data 3.** Regeneration lengths measured with ImageJ/FIJI data for *Figure 3—figure supplement 1E*.

the involvement of OCM and its offshoot pathways in the increased neuronal regeneration in *ogt-1* mutant animals.

## Functional OCM and SSP are indispensable for enhanced regeneration in *ogt-1* animals

Following the result of our gene expression analysis we sought to functionally validate the importance of the OCM and related pathways in neuronal regeneration in *ogt-1* worms. We first focused on the SSP as it metabolically connects glycolysis with OCM (*Figure 3A*; *Yu et al., 2019b*). NCT502 (MCE HY-117240) is a chemical agent reported to inhibits the mammalian phosphoglycerate dehydrogenase (PHGDH) enzyme, which catalyzes the first and rate-limiting step of serine biosynthesis (*Tabatabaie et al., 2010*; *Zogg, 2014*; *Pacold et al., 2016*). Applied to *C. elegans*, NCT502 abrogated the effect of *ogt-1* mutation on neuronal regeneration but significantly increased the regeneration in WT worms (*Figure 3B*, *Figure 3—source data 1*). In addition, we observe that the supplementation of L-serine, the final product of SSP, which feeds into OCM, rescued the abrogative effect of NCT502 in *ogt-1* (*Figure 3B*, *Figure 3—source data 1*). Previously we found that AKT kinase, *akt-1*, activity, plays an important role in *ogt-1* regeneration, *akt-1* mutation blocked the enhanced regeneration of *ogt-1*, while gain of function *akt-1(++)* phenocopied *ogt-1* effect (*Taub et al., 2018*). Interestingly, NCT502 blocked the enhanced regeneration in *ogt-1(-); akt-1(++)* worms (*Figure 3—figure supplement 1A*, *Figure 3—figure supplement 1—source data 1*); and serine supplementation rescued the enhanced regeneration that is eliminated in *akt-1(-);ogt-1(-)* worms (*Figure 3—figure supplement 1A*, *Figure 3—figure supplement 1—source data 1*). Since NCT502 has not been earlier reported to be used in *C. elegans*, we also tested the effects of blocking SSP using RNAi gene knockdown. In concordance with NCT502 treatment, neuron-specific RNAi against C31C9.2 (termed as *phgdh-1*), the *C. elegans* ortholog of human PHGDH and target of NCT502, abrogated the effects of *ogt-1*-mediated regeneration, and significantly increased the regeneration in WT worms even beyond that of *ogt-1* worms (*Figure 3C*, *Figure 3—source data 1*). Interestingly, systemic RNAi knockdown against C31C9.2 (*phgdh-1*), that is ineffective in neurons, did not alter regeneration levels in *ogt-1* animals suggesting a neuron-specific mechanism. However, it did significantly increase regeneration in WT worms (*Figure 3—figure supplement 1B*, *Figure 3—figure supplement 1—source data 2*). We further measured *pyk-1* activity in WT worms and found that it was significantly enhanced by NCT502 treatment (*Figure 3—figure supplement 1C*) suggesting increased glycolytic activity upon blocking the SSP. Interestingly, we observed equally enhanced *pyk-1* activity in *ogt-1* worms with NCT502 treatment (*Figure 3—figure supplement 1D*). These results demonstrate the importance of the SSP pathway in *ogt-1*-mediated enhanced neuron regeneration but suggest that in wild-type animals the reverse may be true and blocking SSP becomes beneficial.

To test the importance of OCM in *ogt-1*-mediated regeneration directly, we tested mutations of methionine synthase (*metr-1*), an ortholog of the human MTR gene and s-adenosyl methionine synthetase-1 (*sams-1*), an ortholog of human MAT1A and MAT2A genes, in the *ogt-1* background (*ogt-1;metr-1*, and *ogt-1;sams-1*) (*Figure 3D*). Both mutations abrogated the enhanced regeneration in *ogt-1* animals but had no significant effect on WT regeneration (*Figure 3E*, *Figure 3—source data 1*). Methionine is an important metabolite of the OCM cycle and its supplementation increases OCM flux (*Miousse et al., 2017*; *Sanderson et al., 2019*; *Ligthart-Melis et al., 2020*). Methionine supplementation significantly increased the regeneration in WT worms but had no additional effect on *ogt-1* worms (*Figure 3F*, *Figure 3—source data 1*, and *Figure 3—figure supplement 1E*, *Figure 3—figure supplement 1—source data 3*). Nor did it alter the effects of blocking the SSP in either WT or *ogt-1* animals (*Figure 3F*, *Figure 3—source data 1*) which may be in part due to the requirement of serine for normal OCM progression downstream (*Yang and Vousden, 2016*; *Clare et al., 2019*; *Geeraerts et al., 2021*). SAM, a product of SAMS-1 and an important metabolite of OCM, mediates numerous cellular processes including several biosynthetic, post-translational modifications and epigenetic modifications of histones and nucleic acids for regulation of gene expression and metabolism, including glycolysis (*Ducker and Rabinowitz, 2017*; *Clare et al., 2019*). It participates in the Kennedy pathway to synthesize lipid (Phosphatidyl Choline) an important component of the cellular membrane (*Figure 3G*; *Walker, 2017*). Phosphatidyl Choline can alternatively be synthesized from choline. However, we found that choline supplementation in *ogt-1;sams-1* dual mutant failed to rescue the effects of *sams-1* mutation (*Figure 3H*, *Figure 3—source data 1*). Furthermore,

neuron-specific RNAi against phosphoethanolamine methyl transferase (*pmt-1*), involved in phosphatidyl choline biosynthesis from SAM, did not reduce *ogt-1*-mediated regeneration, although it did enhance the regeneration in WT worms (*Figure 3H*, *Figure 3—source data 1*). SAM also acts as a methyl doner for transmethylation reactions including histone modification. To test if the epigenetic modification of histones by histone methyltransferases play any role in *ogt-1* enhanced regeneration, we knocked down several reported H3K4 methyltransferase with known effects on H3K4 methylation and/or neuronal regeneration including *set-2, set-11, set-16,* and *set-17* (*Walker et al., 2011*; *Wilson et al., 2020*). Knocking down these methyltransferases had no significant effect on *ogt-1*-mediated enhanced regeneration but significantly increased regeneration in WT worms (*Figure 3I*, *Figure 3—source data 1*). RNA-seq analysis also showed that DNA methylases (*damt-1*) and demethylase (*nmad-1*) as well as *pmt-1/pmt-2*, required for Phosphatidyl Choline synthesis from SAM, were all relatively downregulated while OCM genes were relatively upregulated in neuronal tissue in *ogt-1* animals (*Figure 3—figure supplement 1F*). Thus, while the functional OCM pathway mediated by MERT-1 and SAMS-1 is essential for *ogt-1*-mediated enhanced regeneration, these results suggest that it does not act through either lipogenesis or transmethylation pathways involved in epigenetic regulation.

## The TSP, an offshoot of OCM, is critical for enhanced neuronal regeneration in *ogt-1* animals

Our gene expression analysis revealed that OCM-related pathways such as glutathione and SAM metabolism are highly altered in *ogt-1* worms. We, therefore, tested the importance of the transsulfuration pathway in *ogt-1*-mediated regeneration (*Figure 4A*). The transsulfuration pathway involves cysteine and cystathionine metabolism that is utilized in glutathione synthesis important for oxidative stress maintenance in neurons (*Vitvitsky et al., 2006*; *Sbodio et al., 2019*). Performing neuron-specific RNAi against glutathione synthetase (*gss-1*), an ortholog of GSS, we detected no effect on the enhanced regeneration in the *ogt-1* mutant background but significantly increased regeneration in WT (*Figure 4B*, *Figure 4—source data 1*). In a complimentary manor, supplementation with L-Glutathione (GHS) significantly decreased regeneration in *ogt-1* worms but had no effect on WT worms (*Figure 4B*, *Figure 4—source data 1*). By contrast, supplementation with L-cystathionine had no detectable effect on regeneration in *ogt-1* worms or WT (*Figure 4C*, *Figure 4—source data 1*) but rescued the effect of blocking SSP with NCT502 in *ogt-1* worms (*Figure 4C*, *Figure 4—source data 1*). These observations suggest that while the transsulfuration pathway is functionally involved in *ogt-1*-mediated enhanced regeneration it is not through glutathione synthesis.

Cystathionine can be further metabolized into succinyl-CoA or acetyl-CoA through either the vitamin B12 dependent canonical pathway or the vitamin B12 independent shunt pathways, respectively (*Watson et al., 2016*; *Giese et al., 2020*). Succinyl-CoA or acetyl-CoA can be further used for different metabolic processes or can enter the Krebs Cycle to produce ATP. Our neuronal cell-specific RNA-seq analysis revealed that genes involved in OCM (*metr-1, sams-1, folr-1, mthf-1, etc.*) (*Figure 3—figure supplement 1F*), transsulfuration (*cth-1*) (*Figure 4—figure supplement 1A*), and the downstream vitamin B12 independent shunt pathway (*acdh-1, ech-6, hach-1, hphd-1,* and *alh-8*) were relatively upregulated (*Figure 4—figure supplement 1A*) in *ogt-1* animals, while genes involved in the vitamin B12 dependent canonical pathway (*pcca-1, pccb-1, mce-1,* and *mmcm-1*) were down regulated (*Figure 4—figure supplement 1A*). Likewise, performing qRT-PCR analysis against genes in these pathways, we found that genes involved in TSP (*cth-1, cht-2*) and in the vitamin B12 independent shunt pathway showed unidirectional upregulated expression in *ogt-1* (*Figure 4D*), while genes involved in the canonical vitamin B12 dependent pathway showed no clear trend in differential expression (*Figure 4D*).

To test the role of cystathionine metabolism through shunt and canonical pathways directly in neuronal regeneration, we generated double mutants with acyl-CoA dehydrogenase (*acdh-1*) that mediates the vitamin B12 independent shunt pathway, *ogt-1;acdh-1*, and methylmalonyl-CoA epimerase (*mce-1*) that mediates the vitamin B12 dependent canonical pathway *ogt-1;mce-1* (*Figure 4E*). The *mce-1* mutation had no effect on regeneration in either WT or the *ogt-1* background (*ogt-1;mce-1*) (*Figure 4F*, *Figure 4—source data 1*). However, while *acdh-1* mutation had no effect on WT regeneration, it selectively eliminated the enhanced regeneration of the *ogt-1* background (*ogt-1;cdh-1*) (*Figure 4F*, *Figure 4—source data 1*). These results were recapitulated using neuron-specific RNAi knockdown against *acdh-1* and *mce-1* in the *ogt-1* background (*Figure 4—figure supplement 1B*,

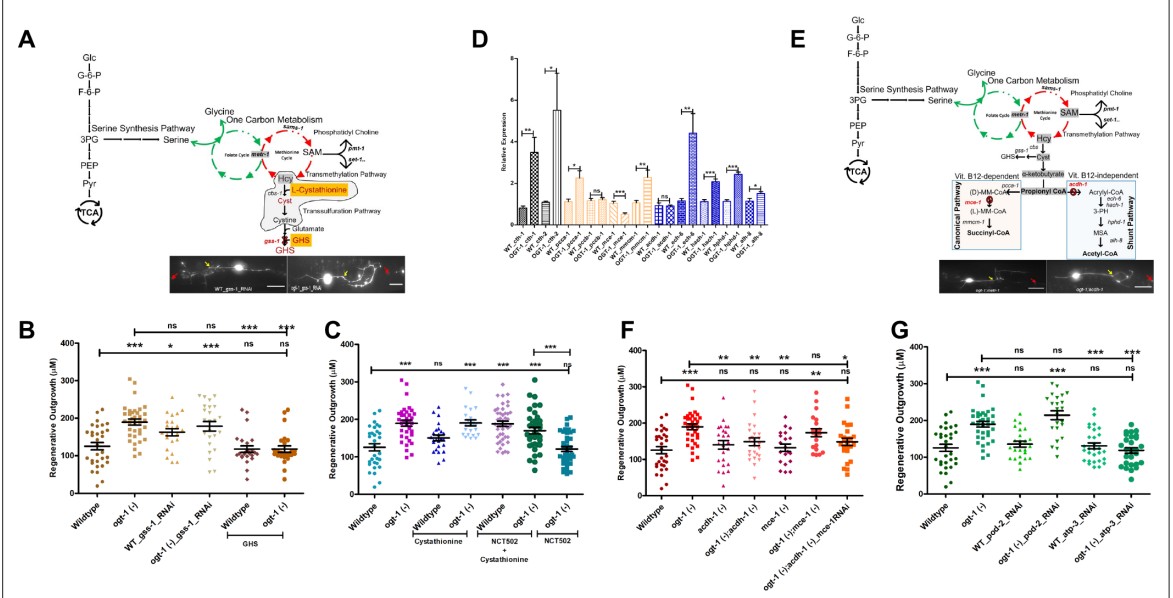

**Figure 4.** The transsulfuration pathway (TSP) leading to acetyl-CoA production mediates enhanced regeneration in O-GlcNAc transferase (*ogt-1*) animals. (**A**) Schematic representation of the transsulfuration pathway (TSP; shaded area) branch of OCM, along with supplementation with TSP metabolites L-cystathionine, Glutathione, and neuron-specific RNAi against Glutathione synthetase (*gss*-1) with its effect on 24 hr neuron regenerating neuron (representative images) (yellow arrow indicates the proximal and red arrow indicated distal point of injury). (**B**) Effects of GHS supplementation and neuronal RNAi knockdown against *gss-1* on neuronal regeneration in wild-type (WT) and *ogt-1* worms. (**C**) Effects of L-cystathionine supplementation on neuronal regeneration in WT and *ogt-1* worms, with or without SSP blocking by NCT502. (**D**) qRT-PCR of selected genes involved in transsulfuration (*cth-1* and *cth-2*), as well as the related downstream vitamin B12 dependent canonical pathways (*pcca-1, pccb-1, mce-1, and mmcm-1*) and the vitamin B12 independent Shunt pathway (*acdh-1, ech-6, hach-1, hphd-1, and alh-8*). (**E**) Schematic representation of the TSP metabolites L-Cystathionine metabolism into succinyl-CoA and acetyl-CoA *via* canonical and shunt pathway, respectively and genes involved with indicated mutants (*acdh-1* and *mce-1*) used in the study, along with a representative regenerating neuron image (yellow arrow indicates the proximal and red arrow indicated distal point of injury). (**F**) Effect of *acdh-1* and *mce-1* mutation in WT and *ogt-1* background on neuronal regeneration. (**G**) Effect of blocking lipid synthesis from acetyl CoA and ATP production on regeneration in WT and *ogt-1*. scale bar = ~10 μM, all data shown in ± SEM, analytical methods student t-test and one-way ANOVA were used *p <0.05, **p <0.01, ***p <0.001 .

The online version of this article includes the following source data and figure supplement(s) for figure 4:

**Source data 1.** Regeneration lengths measured with ImageJ/FIJI data for *Figure 4*.

**Figure supplement 1.** Expression patter important genes from TSP in Neuronal RNAseq data and effect of gene knock down of TSP and lipid metabolism.

**Figure supplement 1—source data 1.** Regeneration lengths measured with ImageJ/FIJI data for *Figure 4—figure supplement 1B*.

**Figure supplement 1—source data 2.** Regeneration lengths measured with ImageJ/FIJI data for *Figure 4—figure supplement 1C*.

*Figure 4—figure supplement 1—source data 1*). Neuron-specific RNAi against *mce-1* in the *ogt-1;acdh-1* double mutant had no observable effect (*Figure 4F*, *Figure 4—source data 1*). The *acdh-1*-mediated shunt pathway is involved in the production of acetyl CoA from L-Cystathionine which is used for several processes including lipid synthesis and/or ATP production. Thus, we tested if lipid synthesis plays a role by neuron-specific RNAi against *pod-2* (acetyl-CoA carboxylase), an ortholog of human ACACA (acetyl-CoA carboxylase alpha), that is important for lipid synthesis from acetyl CoA, but found it had no effect on either WT or *ogt-1* regeneration (*Figure 4G*, *Figure 4—source data 1*). In contrast, the enhanced regeneration in *ogt-1* worms was clearly blocked by neuron-specific RNAi against *atp-3* RNAi that reduces cellular ATP production, (as described above earlier *Figure 1C* and *Figure 4G*, *Figure 4—source data 1*). Since perturbation in O-GlcNAc signaling is known to affect lipid metabolism (*Lockridge and Hanover, 2022*) and lipid metabolism is a significant source of energy in neurons (*Tracey et al., 2018*), we investigated if ATP generation *via* beta-oxidation of lipids plays a role. However, neuron-specific RNAi against, *acs-2*, acyl-CoA synthetase family member 2 (ortholog of human ACSF2), or *cpt-2*, carnitine palmitoyl transferase (ortholog of human CPT1A, carnitine palmitoyl transferase 1 A and CPT1C, carnitine palmitoyl transferase 1 C) did not affect enhanced regeneration in *ogt-1* animals (*Figure 4—figure supplement 1C*, *Figure 4—figure supplement*

*1—source data 2*). While RNAi knockdown does suffer from gene specific variability, both *acs-2* and *cpt-2* are important for lipid beta-oxidation, suggesting that the process is not fundamental in *ogt-1* regeneration. Regardless, in combination with gene expression analysis, these results help to further define the pathway of *ogt-1* regeneration to specifically involve acetyl CoA production by cystathionine metabolism through the vitamin B12 independent shunt pathway.

## Discussion

In order to initiate and sustain the energetically demanding growth state required for effective regeneration there must be sufficient modulation of the underlying molecular and metabolic processes within the damaged neuron (*He and Jin, 2016*). Numerous studies have focused on the molecular and genetic mechanisms involved in axonal regeneration (*Sun et al., 2014*; *Chisholm et al., 2016*; *Chung et al., 2016*). Yet the role of metabolic pathways is relatively less explored, despite its clear role in determining regenerative capacity (*Taub et al., 2018*; *Li et al., 2020*; *Yang et al., 2020*). Previously, our group demonstrated that genetically altered O-GlcNAc levels can substantially enhance neuronal regeneration through modulation of the neuronal metabolic response (*Taub et al., 2018*). Exploiting the genetic and optical accessibility of *C. elegans*, we demonstrated that a reduction of O-GlcNAc levels (*via ogt-1* deletion mutation), a proxy for the metabolic deficit, supports increased regenerative capacity (*Taub et al., 2018*). In this study, we verified these effects in the *ogt-1*(deletion) mutation as before (*Figure 1B*) and found similar results in the catalytically dead mutant allele (OG1135) (*Figure 1B*), demonstrating that the lack of OGT-1 enzymatic activity is important as opposed to a possible non-catalytic function (*Konzman et al., 2022*; *Pravata et al., 2019*). Earlier, we found that disruption of key elements of the glycolytic pathway selectively eliminates the enhanced regeneration of the *ogt-1* mutant (*Taub et al., 2018*). Glycolysis is a key energy source for neurons, particularly under energy-limiting conditions (*Jang et al., 2016*) and in developing neurons that foster high axonal growth rates (*Han et al., 2016*; *Zheng et al., 2016*; *Han et al., 2020*). We further verified this by neuron-specific RNAi knockdown of *atp-3*, which significantly reduces cellular ATP levels (*Soto et al., 2020*) and blocks the enhance regeneration in *ogt-1* animals (*Figure 1D*). While our previous work established that the early steps of neuronal glycolysis are a key component of enhanced axonal regeneration following injury in *ogt-1* worms (*Taub et al., 2018*), key questions remained as to the specific metabolic pathways that are amended and involved to support regeneration.

Our results here indicate that a complex metabolic pathway beyond that of canonical glycolysis is involved (*Figure 5*). In our earlier study, we demonstrated the importance of early glycolytic enzymes (*pfk-1.1* and *pgk-3*) in the *ogt-1* effect (*Taub et al., 2018*). However, we found here that this does not extend to the complete glycolytic pathway as neuron-specific disruption of pyruvate kinase (*pyk-1*), which catalyzes the final step of glycolysis to produce pyruvate, had no effect on regeneration in *ogt-1* (*Figure 1D*). This is in accordance with the reported effects of O-GlcNAcylation on these enzymes. High O-GlcNAcylation decreases *pfk-1.1* function (*Bacigalupa et al., 2018*). Despite the fact that high O-GlcNAcylation also destabilizes the pyruvate kinase, PKM1/2, complex (*Wang et al., 2017*), reports show that inhibition of *ogt-1* results in low pyruvate kinase expression and cellular activity (*Yu et al., 2019a*). The *ogt-1* mutation, which reduces O-GlcNAcylation, is therefore expected to increase *pfk-1.1*, and reduce *pyk-1*, activity, respectively, which agrees with their measured importance in *ogt-1* neuron regeneration.

As these results indicate that the increased regeneration in *ogt-1* mutants does not entail direct ATP production in the TCA cycle of canonical glycolysis, we further adopted an unbiased approach performing genome-wide gene expression analysis to identify additional pathways involved. Through GO and KEGG pathway classification analysis of RNA-seq data from wild-type and *ogt-1* mutant animals, we identified several metabolic pathways altered in both whole animals and FACs sorted neuron samples (*Figure 2B–E*, *Figure 2—source data 1*, and *Figure 2—figure supplement 2C–D*, *Figure 2—source data 2*). In addition to numerous genes and cellular processes with known roles in regeneration such as amino acid, nucleotide metabolism, lipid synthesis, methylation, and glycolysis (*Ducker and Rabinowitz, 2017*; *Clare et al., 2019*), our analysis further identified metabolic processes including glutathione and s-adenosyl methionine (SAM) metabolism, energy metabolism, and ATP synthesis that were significantly enriched in the *ogt-1* background. This pathway enrichment analysis indicates the involvement of OCM and its associated pathways in enhanced regeneration in *ogt-1* animals (*Figure 2B–F* and *Figure 2—figure supplement 2A–B*). These results were further

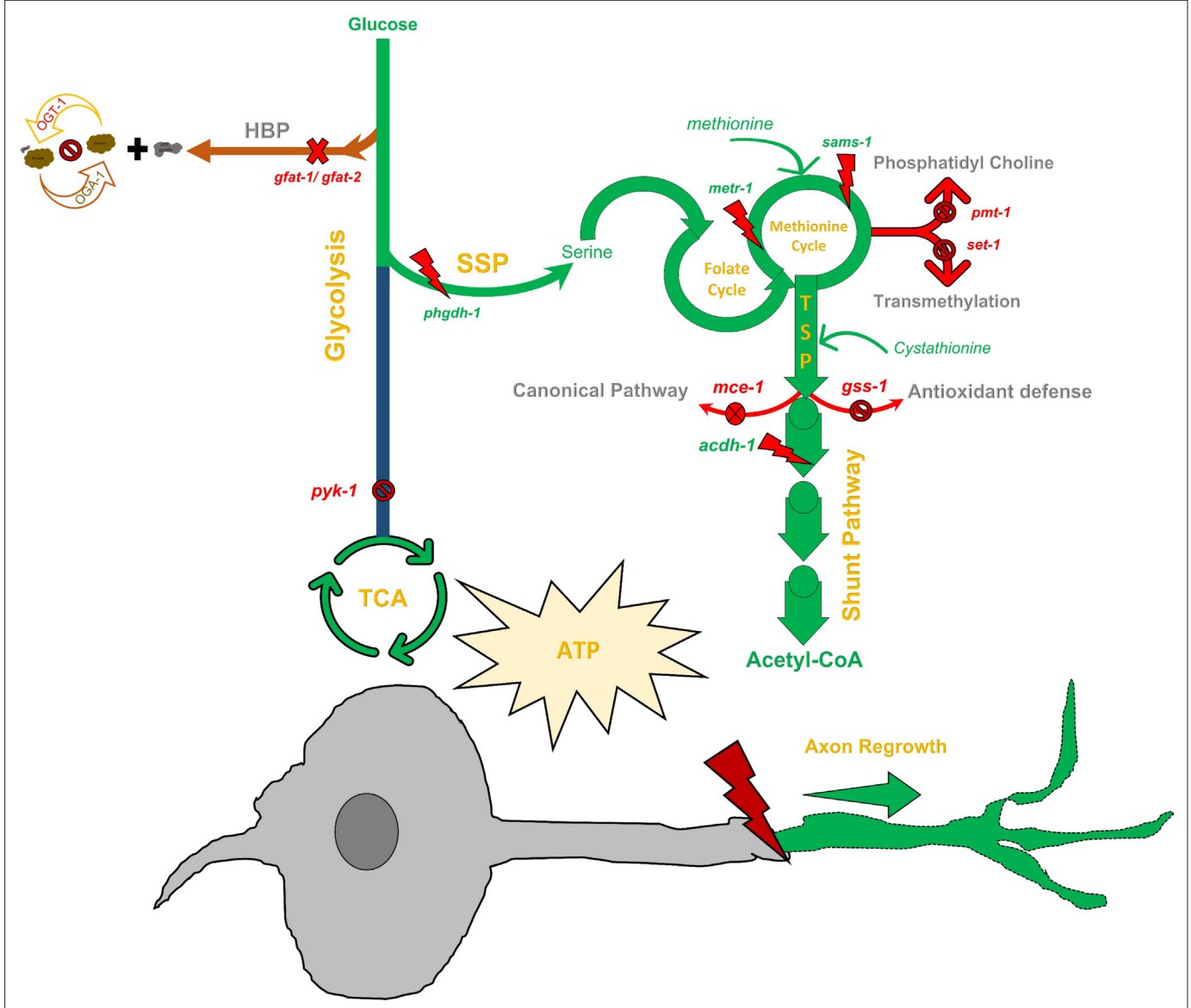

**Figure 5.** The metabolic pathway essential for enhanced neuronal regeneration in O-GlcNAc transferase (*ogt-1*) animals. A detailed schematic of the metabolic pathway essential for the enhanced regeneration in *ogt-1* animals with the tested genes, metabolite supplementations, and pharmacological treatments indicated. As highlighted in green, *ogt-1* mutations required metabolic pathways apart from glycolysis including one-carbon metabolism (OCM), the SPP, the transsulfuration pathway (TSP), and the vitamin B12 independent shunt pathway to support enhanced regeneration. Dispensable metabolic branches are shown in red. Different genetic manipulations (RNAi and mutants) in respective pathways are mentioned at respective places along with methionine and cystathionine metabolites supplementation. Details are mentioned in the main manuscript text.

confirmed *via* specific gene expression analysis using qRT-PCR (*Figure 2G* and *Figure 4D*) and indicate the importance of OCM and the TSP as key metabolic pathways altered by the *ogt-1* mutation (*Figure 2D–F*).

OCM is involved in a wide array of cellular processes including nucleotide biosynthesis (purines and thymidine), amino acid homeostasis (glycine, serine, and methionine), epigenetic maintenance (nucleic acid and histone methylation), and redox defense (*Ducker and Rabinowitz, 2017*). Enhanced glycolysis drives OCM through the SSP (*Locasale, 2013*; *Yu et al., 2019b*) that is known to be involved in several neuronal conditions including, neuronal growth, neural tube defect, and Alzheimer's disease (*Coppede, 2010*; *Bonvento and Bolaños, 2021*; *Lionaki et al., 2022*). Through a combination of genetic manipulation, pharmacological treatment, and metabolic supplementation in our *C. elegans* neuronal regeneration assays, we have determined the specific metabolic pathway by

which OCM contributes to the enhanced regeneration in the *ogt-1* mutant. The complete pathway is illustrated in green in *Figure 5*. We found that metabolic flux diverted from the early steps of glycolysis towards OCM through SSP is crucial, which is in agreement with earlier reports where enhanced glycolysis diverts metabolic flux towards OCM through SSP (*Yu et al., 2019b*). This was most dramatically illustrated by the reduction in regeneration from pharmacological, or genetic, disruption of *phgdh-1* (*C31C9.2*, ortholog of human *PHGDH*), a key element of the SSP. The role of the SSP was further confirm by serine supplementation in the *akt-1* and *ogt-1* double mutant (*ogt-1;akt-1*), which restored the enhanced *ogt-1* regeneration blocked by the *akt-1* mutation (*Figure 3—figure supplement 1A*). These results are in agreement with earlier metabolomic findings that enhanced glycolysis (*Yu et al., 2019b*) and/or knock down of PMK1/2 (mammalian ortholog of *pyk-1*) diverts metabolic flux toward serine synthesis pathway to sustain cellular metabolic requirements (*Yu et al., 2019a*). Here, we have focused our study on SSP, OCM, and TSP because the pathway analysis of our RNA-seq data suggested they were most affected in *ogt-1* animals. However, it is likely that additional metabolic pathways associated with glycolysis, such as the pentose phosphate pathway (PPP, elements of which are dynamically O-GlcNAcylated in response to hypoxic stress *Rao et al., 2015*), are also affected by *ogt-1* mutation and *pyk-1* knock down. Further investigation of these additional pathways should prove beneficial in the future.

Although OCM is involved in both lipogenesis and DNA transmethylation (*Kersten, 2001*; *Yu et al., 2019b*) that could potentially play significant roles in increasing neuron regeneration (*Iskandar et al., 2010*), we found that the regeneration effects of *ogt-1* were primarily dependent on L-cystathionine metabolism *via* the downstream TSP (*Figure 4C*). The TSP is influenced by OCM and its metabolites and has been reported to play an important role in neurodegenerative diseases and ATP production (*Giese et al., 2020*; *Lam et al., 2021*). We found that cystathionine supplementation rescued the prohibitory effects of blocking the SSP pathway in the *ogt-1* background (*Figure 4C*). Testing branches of the TSP, we found that only the vitamin B12 independent shunt pathway was required, *via* Acyl CoA dehydrogenase (*acdh-1*), for *ogt-1*-mediated enhanced regeneration. The shunt pathway generates Acetyl-CoA that will drive ATP production through the Kreb's cycle ultimately bringing the metabolic consequences of *ogt-1* back to cellular energy production and utilization as we demonstrated in Taub et al., Though, we observed a significant decrease in ATP levels (*Figure 1F*, *Figure 1—figure supplement 1C*) and no difference in ATP utilization (*Figure 1—figure supplement 1D*) in *ogt-1* animals, these observations were either in whole worms or in nonneuronal tissues rather than neuron-specific. Indeed, the down regulation of *pyk-1* from *ogt-1* inhibition has been associated with total reduced ATP levels previously (*Dey et al., 2019*). Regardless, our work here has now deciphered a specific metabolic pathway through which the enhanced regenerative effect of *ogt-1* occurs.

While the *ogt-1* mutant rewires metabolic flux through a specific pathway to support and sustain enhance regeneration, we also discovered several additional conditions where restriction or diversion of metabolic flux in wild-type animals has similar beneficial effects. For instance, the HBS pathway nominally shunts off ~5% of glycolytic flux (*Marshall et al., 1991*; *Bond and Hanover, 2015*). We found that blocking the HBS pathway through RNAi against *gfat-1* and *gfat-2* (*Yi et al., 2012*; *Jóźwiak and Forma, 2014*; *Kim et al., 2018*), appears to divert metabolic flux towards glycolysis and results in enhanced regeneration in WT animals similar to that of *ogt-1* (*Figure 1C*). Likewise, *pyk-1* knockdown increases regeneration in WT and is known to divert metabolic flux toward the SSP (*Yu et al., 2019a*). Within OCM, we found that transmethylation pathways required for epigenetic modifications and phospholipid synthesis were not essential for the enhanced regeneration in *ogt-1* animals but that blocking histone methyl transferases in WT animals increased regeneration (*Figure 3I*). In addition, supplementation in wild type with the metabolite, L-methionine (product of *metr-1*), which increases OCM, phenocopied the enhance regeneration of the *ogt-1* mutant (*Figure 3F*) as did blocking neuronal glutathione synthesis within the TSP (*gss-1* RNAi) (*Figure 4B*). While in the above instances restriction or enhancement of specific metabolic steps could be augmenting the same pathway utilized in *ogt-1* regeneration, in other cases clearly alternative pathways are at work. For example, pharmacologically (NCT502 treatment) or genetically (*phgdh-1* knock down) blocking SSP which restricts the *ogt-1* regeneration pathway effectively increases regeneration in WT. This effect is possibly due to increased metabolic flux through glycolysis, as we observed increased activity of *pyk-1* after NCT502 treatment (*Figure 3—figure supplement 1C*). Likewise, we had previously found that mutation of the O-GlcNAcase, *oga-1*, which increases O-GlcNAc levels, also increased neuron

regeneration in *C. elegans,* but did so through an independent pathway of enhanced mitochondrial stress response (*Taub et al., 2018*).

Thus, within the complex web of cellular metabolism and energy production there appears to be numerous pathways for metabolite utilization that are beneficial for neuron regeneration. Here, employing genetic tools, we have defined a specific array of metabolic pathways (glycolysis, SSP, OCM, and TSP) through which the *ogt-1* mutation diverts metabolic flux to increase neuronal regeneration. It is important to emphasize the accessibility of these metabolic effects to pharmacological treatment and/or metabolite supplements. For example, we previously demonstrated increased regeneration in wild-type animals with glucose supplementation (*Taub et al., 2018*). Here, we find similar effects with L-methionine supplementation or treatment with the SSP blocking agent NCT502 in wild-type animals. Nutrient supplements and metabolic drug targets have been employed in neurotherapeutic treatments and prevention in numerous contexts including neuronal developmental defects (*Greene et al., 2017*; *Businaro et al., 2021*; *Wu et al., 2022*) and age-associated neurodegenerative diseases (*Stempler et al., 2014*; *Businaro et al., 2021*). While our current study has specifically focused on the neuronal regeneration of the mechanosensory neurons in *C. elegans*, O-GlcNAc signaling has been widely implicated in regulating various neuronal cellular processes in higher organisms. This includes axon growth, synaptic plasticity, and neurite outgrowth (*Tian et al., 2023*; *Mutalik and Gupton, 2021*) and has been linked to the modulation of gene expression during neuronal repair in higher animals including mammals (*Lee et al., 2021*; *Su and Schwarz, 2017*). These studies suggest that functions of O-GlcNAc signaling are conserved throughout the nervous system and across species. Future work will be required to determine to what degree this holds true specifically in the context of neuronal regeneration. Regardless, our work here demonstrates the role of OCM, SSP, and TSP metabolic pathways in the increased regenerative capacity of a damaged neuron in *ogt-1 C. elegans* and highlights the potential power for such metabolic targets in the treatment of neuronal injury.

## Materials and methods
### Reagents and resources
Further information and requests for resources, data, and reagents should be directed to and will be fulfilled by the Lead Contact, Christopher V. Gabel (cvgabel@bu.edu).

### Experimental model and subject details
All *C. elegans* strains were cultured and maintained at 20 °C on nematode growth media (NGM) agar plates seeded with OP50 *E. coli*, unless otherwise noted. Strains were obtained from the *Caenorhabditis* Genetics Consortium (CGC at the University of Minnesota). To visualize the mechanosensory neurons, strains were crossed either into SK4005 (zdis5 [pmec4::GFP]) or *ogt*-1::zdis-5. Strains used are listed in detail in the appendix table at the end of the manuscript. All strains generated by crossing were confirmed by genotyping using primers recommended by the CGC.

### Laser axotomy
In vivo Laser Axotomy was performed with a Ti:Sapphire infrared laser system (Mantis PulseSwitch Laser, Coherent Inc), that generated a 1 mHz train of 100 fs pulses in the near infrared (800 nm), pulse energy of 15–30 nJ/pulse or with a Yb-doped diode-pumped solid-state laser (SpectraPhysics Spirit-1040–4 W) outputting 1040 nm 400 fs pulses at 1 kHz (*Wang et al., 2022*). Axotomy was performed on a Nikon Ti-2000 inverted fluorescent microscope with a Nikon 40X 1.4 NA objective or with a Nikon Plan Apochromat Lambda 1.4 NA, 60 X oil-immersion objective. Neurons were imaged for axotomy and subsequent measurement of regeneration *via* standard wide-field fluorescence of gfp expressed in the targeted neuron. Day 1 adult *C. elegans* were mounted on 5–6% agarose pads and immobilized in a 3–5 μL slurry of polystyrene beads (Polysciences, Polybead Polystyrene, 0.05 μM microsphere, cat#08691–10) and NGM buffer (*Kim et al., 2013*). Axotomy consisted of 3–5 short laser exposures (0.25 s each) resulting in vaporization at the focal point and severing of the targeted axon. The anterior lateral microtubule (ALM) neuron was injured with two targeted cuts. The first cut was made 20 μm from the cell soma and a second cut was made 40–50 μm from the cell soma, creating a 20–30 μm gap. Regeneration lengths were reimaged with a Nikon 40X 1.4 NA objective 24 hr after axotomy, or as otherwise indicated, by placing the animals on a 2% agarose pad with 5 mM sodium

azide. Regeneration lengths were measured by tracing along the new neuron outgrowth both from the proximal ablation point as well as any new backward growth from the soma with ImageJ/FIJI. On an average for each condition, we have performed axotomy and measured regeneration for at least ~20 worms, exact numbers for each experiment are given in the associated supplementary data tables.

## Mechanosensory neuron-specific RNAi feeding

To evaluate the function of specific genes, RNAi gene knockdown was employed following protocols we used previously, Taub et al., These protocols were first confirmed by performing RNAi knockdown against GFP in mechanosensory neurons. Significant reduction in GFP fluorescence was observed equivalent to the results reported in Taub et al, confirming neuron specific gene knockdown. Both the Ahringer and Vidal bacterial RNAi libraries were employed (*Kamath and Ahringer, 2003*; *Rual et al., 2004*). Following standard protocols, bacteria colonies were streaked out on LB agar containing penicillin and grown at 37 °C overnight. The next day, single colonies were selected and grown in 10 mL of LB with Ampicillin overnight at 37 °C. From this subculture, 250 micro-liters (uL) were spread onto RNAi agar plates containing penicillin and 2 mM IPTG. Plates were dried and incubated at room temperature for at least 48 hr before using them for worm culturing. For mechanosensory neuron-specific RNAi gene knockdown, we employed the TU3568 (sid-1(pk3321) him-5(e1490) V; lin-15B(n744) X; uIs71 [(pCFJ90) pmyo-2::mCherry + pmec-18::sid-1]) background (*Calixto et al., 2010*). This strain has RNAi sensitivity specifically in the mechanosensory neurons and is RNAi resistant in all other tissues. TU3568 was crossed into the *ogt-1* mutant background. Following protocols we established in *Taub et al., 2018*, gravid adults were bleached, and embryos were allowed to hatch onto RNAi-bacteria plates. Once the F1 generation reached adulthood, 30–40 gravid adults were picked onto fresh RNAi-bacteria plates and allowed to lay eggs for 3–4 hr. The day 1 adults of the F2 generation was then used for Laser Axotomy and regeneration assays as described above. Animals were rescued on a fresh RNAi plate and cultured until imaging was performed.

## Drug treatments in *C. elegans*

For all chemical reagent and metabolite treatments, the compound was dissolved in NGM agar before being poured into plates. Animals were cultured on treated plates for their lifespan before and after axotomy. Choline 30 mM (Sigma, Cat#: C7017-5G), (*Ding et al., 2015*), L-Methionine 75 µM (Fisher Scientific, Cat#: AC166160025), 5 mM L-Serine (Sigma, cat# S-4500) (*Liu et al., 2019*), L-Cystothionine 50 µM (Sigma, Cat#: C7017-5G, CAS:67-48-1), and L-Glutathione reduced 100 µM (Cat#: G4251-50G), were dissolved in molecular grade water (Fisher Scientific, Cat#: R91450001G, CAS: 7732-18-5) at required stock concentrations (*Ellwood et al., 2022*). The phgdh (C31C9.2) inhibitor N-(4,6-dimethylpyridin-2-yl)–4-[5-(trifluoromethyl)yridine-2-yl]piperazine-1-carbothioamid (NCT502) (MedChemExpress, Cat#: HY-117240) was initially dissolved in DMSO and diluted in ddH2O to use at a concentration of 25 µM in NGM plates (*Pacold et al., 2016*). Note: We have shown previously that DMSO does not affect regeneration (*Taub et al., 2018*). All the drugs, chemicals, and kits used are listed in the appendix table at the end of the manuscript.

## qRT-PCR

To evaluate the expression levels of candidate genes in wild-type and *ogt-1* animals we performed qRT-PCR. Day 1 adult *C. elegans* were lysed in 0.5% SDS, 5% b-ME, 10 mM EDTA, 10 mM Tris-HCl pH 7.4, 0.5 mg/ml Proteinase K, then RNA was purified with Tri-Reagent (Sigma). DNAse I treatment (NEB M03035) of 2–3 ug RNA followed by cDNA conversion using High-Capacity cDNA Reverse Transcription Kit (Thermo Fisher Scientific cat#4368814). qRT-PCR was performed in biological triplicate with three technical triplicates for each condition using Real-Time PCR Quantstudio 12 K Flex qPCR System and Fast SYBR Green Master Mix (Thermo Fisher, 4385617). Relative transcript abundance was determined by using the DDCt method and normalized to *act-1* mRNA expression levels as a control. Primers are listed in the appendix table at the end of the manuscript.

## Neuronal cell isolation from adult animals using FACs

To isolate neuronal cells from Day 1 adult worms we utilized the protocol developed and described earlier (*Zhang et al., 2011*; *Kaletsky et al., 2016*). In brief WT (*unc-119*::GFP) and *ogt-1*

(*ogt-1::unc-119*::GFP) worms expressing GFP in all neurons were generated by crossing WT or OGT-1 worms with otIs45 [*unc-119*::GFP]. Synchronized day 1 adult worms were washed (3 X) with s-basal buffer to remove excess bacteria. The packed worm volume (250–350 µl) was washed twice with 500 µl lysis buffer (200 mM DTT, 0.25% SDS, 20 mM HEPES pH 8.0, 3% sucrose) and resuspended in 1000 µl lysis buffer. Worms were incubated in lysis buffer with intermittent gentle tapping for 10 min at room temperature. The pellet was washed 6 X with s-basal and resuspended in 20 mg/ml pronase solution from Streptomyces griseus (Sigma- Aldrich, SKU# 10165921001). Worms were incubated at room temperature (15–20 min) with periodic mechanical disruption by pipetting at every 2 min intervals. When most worm bodies were dissociated, leaving only small debris and eggs (as observed under a dissecting microscope), dissolved whole worm tissues were filtered to remove eggs and single cells were pelleted down at 4 K RPM for 20 min at 4 °C. The pellets were resuspended in ice-cold PBS buffer containing 2% fetal bovine serum (Gibco). The resulting dissociated cell suspension was subjected to Fluorescence-activated cell sorting (FACs) to isolate GFP labeled neurons (*Figure 2— figure supplement 1A*).

## Expression profiling by RNA-seq

Gene expression patterns in WT and *ogt-1* mutants were measured by RNA-seq analysis from RNA extracted from both, day 1 adult, whole animal and FACs sorted neuronal cells. RNA from FACS-sorted neurons were extracted using the Direct-zol RNA Miniprep Plus Kit (Zymo Research, R2070). RNA from whole animals was extracted manually by lysing day 1 adult *C. elegans* in 0.5% SDS, 5% b-ME, 10 mM EDTA, 10 mM Tris-HCl pH 7.4, 0.5 mg/ml Proteinase K, then RNA was purified with Tri-Reagent (Sigma cat# T9424-25ML). Isolated RNA was purified by RNAeasy columns (QIAGEN, Cat#74034), and the quality of RNA was evaluated with the 2100 bioanalyzer (Agilent) before library generation for the RNA-seq experiments. RNA-seq experiments were not randomized, nor results blinded, as all analysis is fully automated and unbiased. For whole-worm and neuron-specific RNA sequencing of adult animals N=2 biological replicates were used. No statistical methods were used to predetermine sample size (*Kaletsky et al., 2016*).

For whole-body RNA-seq analysis we acquired DNBseq RNA sequencing services from BGI Global (https://gtech.bgi.com/bgi/home). Total RNA-seq and data analysis were performed by using BGI Global inhouse developed sequencing methods and data analysis. In brief, transcriptome libraries were generated using the library conversion kit before sequencing was performed on the DNBseq platform. For each library, 10 ng library was used to incorporate a 5′ phosphorylation, on the forward strand only, using polymerase chain reaction (PCR). Purified PCR product with 5′ phosphorylation was then denatured and mixed with an oligonucleotide 'splint' that is homologous to the P5 and P7 adapter regions of the library to generate a ssDNA circle. A DNA ligation step was then performed to create a complete ssDNA circle of the forward strand, followed by an exonuclease digestion step to remove single-stranded non-circularized DNA molecules. Circular ssDNA molecules were then further subjected to Rolling Circle Amplification (RCA) to generate DNA Nanoballs (DNB) containing 300–500 copies of the libraries. Each DNB library was then drawn through a flow cell ready for sequencing using the DNBseq platform to generate 30 M clean reads per sample. FASTQ files were generated locally at sequencing performed by BGI. After data cleaning, processing includes removing adaptors, contamination, and low-quality reads. Bowtie2 was used to map the clean reads to the reference gene sequence (transcriptome), and then RSEM was used to calculate the gene expression level of each sample. The DEseq2 method was used to detect DEGs.

For neuron-specific RNA-seq analysis we employed the Illumina NextSeq 2000 RNA sequencing services from 'The Boston University Microarray & Sequencing Resource' (https://www.bumc.bu.edu/microarray/). RNA isolated from FACs-sorted neuronal cells were subjected to quality control assessment using a bioanalyzer (Aligent). mRNA enrichment, library preparation and quality assessments were performed according to manufacturer protocols (Illumina). Sequencing was performed on the Illumina NextSeq 2000 System using the NextSeq 2000, P2 Reagent Kit (100 cycles) with sequencing read length 50 × 50 paired-end. Sequencing data were assessed for the quality of each sample using **FastQC** (https://www.bioinformatics.babraham.ac.uk/projects/fastqc/), and **RSeQC** (https://rseqc.sourceforge.net/). Each sample was aligned to the genome using **STAR** (https://github.com/alex-dobin/STAR; *Dobin, 2023*), and **SAMtools** (https://samtools.sourceforge.net/) was used to count proper pairs of reads aligning to mitochondrial or ribosomal RNA. We confirmed *ogt-1* mutation by

PCR genotyping (note: sequence alignment to the *C. elegans* genome using **STAR** was unable to confirm it). The subread package: high-performance read alignment, quantification, and mutation discovery **featureCounts** (https://subread.sourceforge.net/) was used for alignment of proper read pairs unique to non-mitochondrial Ensemble Genes. As a control, all reads were also aligned to the GFP sequence, which indicated that all samples were GFP-positive as expected. To identify genes whose expression changes significantly between genotypes, a one-way analysis of variance (ANOVA) was performed using a likelihood ratio test to obtain a p-value for each gene. Benjamini-Hochberg false discovery rate (FDR) correction was applied to obtain FDR-corrected p-values (q-values), which represent the probability that a given result is a false positive based on the overall distribution of p-values. The FDR q-value was also recomputed after removing genes that did not pass the 'independent filtering' step in the DESeq2 package. Wald tests were then performed for each gene between experimental groups to obtain a test statistic and p-value for each gene. FDR correction was then applied, across all genes for which a p-value could be computed for all comparisons and across only those genes that passed expression filtering.

## RNA-seq bioinformatic analysis

The following unbiased enrichment analysis was used to understand whether the differentially expressed gene list identified in the RNA-seq data was significantly enriched in a pathway, molecular function, or particular biological process. **GO** was employed to determine the molecular function, cellular component, and biological process of the differentially expressed genes. All differentially expressed genes where mapped to terms in the Gene Ontology database (http://www.geneontology.org/), the number of genes in each term calculated, and a hypergeometric test applied to identify GO terms that are significantly enriched in candidate genes compared to the background of all genes in the species. In addition, we also utilized the online ShinyGO v0.741: Gene Ontology Enrichment Analysis (http://bioinformatics.sdstate.edu/go74/) to analyze neuronally enriched genes. **KEGG Pathway-based analysis** (q-value ≤0.05) was employed to determine the most important biochemical metabolic and signal transduction pathways significantly enriched in the differentially expressed genes. The differentially expressed gene list was further analyzed for functional annotation of enriched pathways using The **Database for Annotation, Visualization, and Integrated Discovery (DAVID)**. These tools are powered by the comprehensive DAVID Knowledgebase built upon the DAVID Gene concept which pulls together multiple sources of functional annotations. Using the recommended protocol for analysis in the wizard tool of DAVID (https://david.ncifcrf.gov/tools.jsp) we analyzed the pathways and metabolites most affected in neurons of *ogt-1* mutants.

## ATP quantification via the FRET-based ATP sensor

We obtained the worm strain (MS2495) expressing the fluorescence resonance energy transfer (FRET) based ATP sensor (novel**C**lover-**A**TP-**mA**pple fusion protein; **CAmA**) under the *pept-1* promoter expressed in the intestinal cells from Dr. Morris F Moduro lab (*Soto et al., 2020*). Clover is a green fluorescent protein that is excited by blue light (480 nm-510 nm laser) and emits green light (511 nm-530 nm). mApple is a red fluorescent protein that is excited by green light (522 nm-577 nm) and emits red light (580 nm-675 nm). The *ogt-1* mutant was crossed with the ATP sensor strain (MS2495). The anterior gut of day 1 adult worms (control and *ogt-1* mutant) was imaged to measure FRET fluorescence using a 63 x objective on a confocal Zeiss LSM 880 microscope. Following established FRET imaging protocols, a mApple image was acquired first *via* direct excitation (561 nm laser) and emission (594 nm) to assess where the sensor protein was present and establish a baseline measurement. A second image was then obtained using a FRET filter set, i.e., excitation of Clover (488 nm laser), producing green emission (522 nm-577 nm) that excites mApple which is detected as red emission (516 nm) (FRETred). ImageJ was used to quantify the relative FRET pixel intensity (FRETred/baseline) within the region of interest.

## ATP and PPi quantification and *pyk-1* activity assay

Synchronized day 1 adult worms were collected in S-basal buffer and were washed 3 x with s-basal and 1 x in ATP assay buffer (Abcam, Ab83355), followed by sonication on ice in ATP assay buffer using a model 110 V/T Ultrasonic Homogenizer for two cycles of 15 min. Sonicated samples were then centrifuged at 13,000 RPM for 15 min at 4 °C. The supernatant was collected and moved to a fresh microcentrifuge

tube and ATP quantitation was performed with the ATP Assay Kit (Colorimetric/Fluorometric) (Abcam, Ab83355) using a Tecan Infinite M1000 Pro Multi Microplate Reader. ATP was normalized to protein content measured with the BCA Protein Quantification Kit (Abcam, Ab102536). Triplicate technical replicates were performed for each sample; at least three biological samples were assayed for each condition reported. For *pyk-1* activity and pyrophosphate PPi quantification assays, animals were cultured as in ATP quantification assays and animals were sonicated on ice in respective assay buffers (*pky-1* or PPi assay buffer) and activity was recorded using a Tecan Infinite M1000 Pro Multi Microplate Reader. To normalize samples, the BCA Protein Quantification Kit (Abcam, Ab102536) was used.

## Quantification and statistical analysis

Statistical analysis and graph generation was performed with Prism (GraphPad). All data were compared with either WT, *ogt-1* mutant, or RNAi control regeneration data. Data are shown as the mean with error bars representing the standard error of the mean. One-way ANOVA analysis with Dunnett's and *post hoc* Bonferroni's correction was employed for multiple comparisons. When only two groups of data were compared an unpaired t-test was employed. In all cases, $*p<0.05$ $**p<0.01$, $***p<0.001$.

## Acknowledgements

We would like to thank Dr. Amy Walker, UMASS Worcester, MA, USA, Dr. Morris Maduro, UC Riverside, and The *C. elegans* Genetics Center provided many of the strains and Boston University Core Dr. Tilton, Brian Richard (FACs); Dr. Yuriy Alekseyev (RNA sequencing and data analysis), Dr. Trinkaus-Randall (confocal Zeiss LSM 880 microscopy), and Dr. Lyn and Au, Matthew Bo (qRT-PCR) facilities for maintaining and making available different instruments for use. Dr. Walker provided *sams-1* worm, Dr. Maduro provided ATP expressing strain. Dr. Danial Taub provided feedback and suggestions on the manuscript. Funding was provided by the Massachusetts Spinal Cord Injury Cure Research Program, INTF3110HH2191525007, from the Massachusetts Department of Public Health to CVG and NIH R56NS128413 to SHC.

## Additional information

### Funding

| Funder | Grant reference number | Author |
| --- | --- | --- |
| Massachusetts Department of Public Health | Massachusetts Spinal Cord Injury Cure Research Program, INTF3110HH2191525007 | Dilip Kumar Yadav Andrew C Chang Christopher V Gabel |
| National Institutes of Health | R56NS128413 | Noa WF Grooms Samuel H Chung |

The funders had no role in study design, data collection and interpretation, or the decision to submit the work for publication.

### Author contributions

Dilip Kumar Yadav, Conceptualization, Data curation, Formal analysis, Investigation, Writing – original draft, Writing – review and editing, Conceived and designed experiments. Performed all the experiments and aided in the analysis of data; Andrew C Chang, Data curation, Investigation, Aided in confocal imaging and analysis; Noa WF Grooms, Investigation, Aided in laser ablation experiments; Samuel H Chung, Funding acquisition, Investigation, Project administration, Provided laser ablation equipment and guidance; Christopher V Gabel, Conceptualization, Funding acquisition, Investigation, Writing – original draft, Project administration, Writing – review and editing, Conceived and designed experiments. Aided in all experiments and analysis of data

### Author ORCIDs

Dilip Kumar Yadav  http://orcid.org/0000-0002-0232-7387
Christopher V Gabel  https://orcid.org/0000-0002-2763-3938

**Decision letter and Author response**
Decision letter https://doi.org/10.7554/eLife.86478.sa1
Author response https://doi.org/10.7554/eLife.86478.sa2

---

## Additional files

### Supplementary files
• MDAR checklist

### Data availability
RNAseq data have been deposited at NCBI under accesses codes PRJNA938796 and PRJNA938805.

The following datasets were generated:

| Author(s) | Year | Dataset title | Dataset URL | Database and Identifier |
|---|---|---|---|---|
| Yadav DK | 2023 | *Caenorhabditis elegans* Raw sequence reads | http://www.ncbi.nlm.nih.gov/bioproject/?term=PRJNA938796 | NCBI BioProject, PRJNA938796 |
| Yadav DK | 2023 | *Caenorhabditis elegans* Neuronal Cells RNA sequence data | http://www.ncbi.nlm.nih.gov/bioproject/?term=PRJNA938805 | NCBI BioProject, PRJNA938805 |

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

# Appendix 1

Key resource table: List of *E. coli* and *C. elegans* strains, Chemicals, Drugs, Kits, and RT-PCR primers used in this study.

**Appendix 1—key resources table**

| Reagent type (species) or resource | Designation | Source or reference | Identifiers | Additional information |
|---|---|---|---|---|
| Strain, strain background (*Escherichia coli*) | *Caenorhabditis* Genetics Centre | Wormbase, *E. coli* OP50 | WBStrain00041969 | Other names: CBb1 |
| Recombinant DNA reagent | Source Bioscience | Vidal and Ahringer RNAi Libraries in HT115 (D3) *E. coli* | | RNAi Library |
| Chemical compound, drug | L-methionine | Fisher Scientific | Cat#: AC166160025 CAS: 63-68-3 | |
| Chemical compound, drug | L-Cystathionine | Sigma | Cat#: C7505-10MG | |
| Chemical compound, drug | L-Methionine | Sigma | Cat#: M-9625 | |
| Chemical compound, drug | L-Serine | Sigma | Cat#: S-4500 | |
| Chemical compound, drug | Choline Chloride | Millipore Sigma | Cat#: C7017-5G, CAS:67-48-1 | |
| Chemical compound, drug | Sodium Chloride (NaCl) | Fisher Bioreagents | Cat#: BP358-1 CAS:7647-14-5 | |
| Chemical compound, drug | β-meracaptoethanol | Fisher Scientific | Cat#: AC125470100 CAS: 60-24-2 | |
| Chemical compound, drug | Trizol | ThermoFisher Scientific | Cat#: 15596018 | |
| Chemical compound, drug | PowerUp SYBR Green Master Mix | Applied Biosystems | Cat#: A25741 | |
| Chemical compound, drug | Agarose | Fisher Bioreagents | Cat#: BP160-500, CAS: 9012-36-6 | |
| Chemical compound, drug | Pronase | Sigma- Aldrich | SKU# 10165921001 | |
| Chemical compound, drug | DMSO (Dimethyl Sulfoxide) | ThermoFisher Scientific | Cat#: 85190, Cas:67-68-5 | |
| Chemical compound, drug | Water, Molecular Grade, Sterile, DEPC Free | Fisher Scientific | CAS: 7732-18-5 Cat#: R91450001G, | |
| Chemical compound, drug | NCT502 | MedChemExpress (MCE) | Cat#: HY-117240 | |
| Chemical compound, drug | Polybead polystyrene | Polysciences | Cat#08691–10 | |
| Commercial assay or kit | ATP Assay Kit (Colorometric/Fluorometric) | Abcam | Ab83355 | |
| Commercial assay or kit | Pyrophosphate Assay Kit (Fluorometric) | Abcam | Ab112155 | |
| Commercial assay or kit | Pyruvate Kinase (PK) Assay Kit (Colorimetric) | Abcam | Ab83432 | |
| Commercial assay or kit | BCA Protein Quantification Kit | Abcam | Ab102536 | |
| Commercial assay or kit | RNAeasy columns | QIAGEN | Cat#74034 | |
| Commercial assay or kit | Direct-zol RNA Miniprep Plus Kit | Zymo Research | Cat# R2070 | |

*Appendix 1 Continued on next page*

Appendix 1 Continued

| Reagent type (species) or resource | Designation | Source or reference | Identifiers | Additional information |
|---|---|---|---|---|
| Strain, strain background (C. elegans) | SK4005 | Taub et. al. | | WT (zdis5 pmec-4::GFP) |
| Strain, strain background (C. elegans) | NA | Taub et. al. | | ogt-1(ok1474)_zdis-5 pmec-4::GFP |
| Strain, strain background (C. elegans) | OG1135 | This study | | Ogt-1 (OG1135)_TU3568_ pmec-4::GFP |
| Strain, strain background (C. elegans) | RB1342 | Taub et. al. | | ogt-1(ok1474) |
| Strain, strain background (C. elegans) | NA | Taub et. al. | | TU3568 (sid-1(pk3321) him-5(e1490) V; lin-15B(n744) X; uIs71[(pCFJ90) pmyo-2::mCherry +pmec-18::sid-1]) |
| Strain, strain background (C. elegans) | NA | Taub et. al. | | ogt-1(ok1474)_TU3568 |
| Strain, strain background (C. elegans) | RB2240 | CGC | | sams-1(ok3033) |
| Strain, strain background (C. elegans) | NA | in this study | | sams-1_zdis-5 |
| Strain, strain background (C. elegans) | NA | in this study | | ogt-1;sams-1_zdis-5 |
| Strain, strain background (C. elegans) | RB755 | CGC | | metr-1(R03D7.1(ok521)) |
| Strain, strain background (C. elegans) | NA | in this study | | metr-1_zdis-5 |
| Strain, strain background (C. elegans) | NA | in this study | | ogt-1;metr-1_zdis-5 |
| Strain, strain background (C. elegans) | VC1011 | CGC | | acdh-1(ok1489) |
| Strain, strain background (C. elegans) | NA | in this study | | acdh-1_zdis-5 |
| Strain, strain background (C. elegans) | NA | in this study | | ogt-1;acdh-1_zdis-5 |
| Strain, strain background (C. elegans) | RB512 | CGC | | mce-1(D2030.5(ok243)) |
| Strain, strain background (C. elegans) | NA | in this study | | mce-1_zdis-5 |
| Strain, strain background (C. elegans) | NA | in this study | | mce-1_TU3568_zdis-5 |
| Strain, strain background (C. elegans) | NA | in this study | | ogt-1;mce-1_TU3568_zdis-5 |

Appendix 1 Continued on next page

*Appendix 1 Continued*

| Reagent type (species) or resource | Designation | Source or reference | Identifiers | Additional information |
|---|---|---|---|---|
| Strain, strain background (*C. elegans*) | CG122 | Taub *et. al.* | | *ogt-1;akt-1*(mg144) |
| Strain, strain background (*C. elegans*) | CG125 | Taub *et. al.* | | *ogt-1;akt-1*(ok525) |
| Strain, strain background (*C. elegans*) | MS2495 | Soto and Rivera *et. al.* | | irls158 (normal ATP sensor, CAmA) |
| Strain, strain background (*C. elegans*) | NA | In this study | | *ogt-1; CAmA* |
| Sequence-based reagent | *folr-1_F* | C17G1.1 | RT-qPCR | GGCTTCCATTGCCGTCATAA |
| Sequence-based reagent | *folr-1_R* | C17G1.1 | RT-qPCR | GCTAACCACTGGCTCACGAT |
| Sequence-based reagent | *metr-1_F* | R03D7.1 | RT-qPCR | CCCGAATCGCAGTTATCCGA |
| Sequence-based reagent | *metr-1_R* | R03D7.1 | RT-qPCR | GAAGCAGCTGGGAGGAATGA |
| Sequence-based reagent | *sams-1_F* | C49F5.1 | RT-qPCR | CACTCACCGACGAAGAGCTT |
| Sequence-based reagent | *sams-1_R* | C49F5.1 | RT-qPCR | GTGACCGAAGTGACCGTTCT |
| Sequence-based reagent | *dmat-1_F* | C18A3.1 | RT-qPCR | ATTGCCGATCCACCATGGTT |
| Sequence-based reagent | *dmat-1_R* | C18A3.1 | RT-qPCR | CCGATTTGTGATCCAGAAAGCA |
| Sequence-based reagent | *nmad-1_F* | F09F7.7 | RT-qPCR | GCACAGTCACAAAGTGGTCG |
| Sequence-based reagent | *nmad-1_R* | F09F7.7 | RT-qPCR | CGTACTCTGGCATTCCGACA |
| Sequence-based reagent | *cth-1_F* | F22B8.6 | RT-qPCR | TCTGATATTATTATGGGAGCCGC |
| Sequence-based reagent | *cth-1_R* | F22B8.6 | RT-qPCR | TGCAGTCATGAGCTTCAAGGA |
| Sequence-based reagent | *cth-2_F* | ZK1127.10 | RT-qPCR | TTGGAGCGGATGTTGTCGTT |
| Sequence-based reagent | *cth-2_R* | ZK1127.10 | RT-qPCR | AGTGAGCTCTCATTCTGATGTGA |
| Sequence-based reagent | *pcca-1_F* | F27D9.5 | RT-qPCR | AAATGGGAGAACAGGCCGTT |
| Sequence-based reagent | *pcca-1_R* | F27D9.5 | RT-qPCR | TGGGTGATTGGAAGTGGGTG |
| Sequence-based reagent | *pccb-1_F* | F52E4.1 | RT-qPCR | AAAGTTTGCTGCTGGATGCC |
| Sequence-based reagent | *pccb-1_R* | F52E4.1 | RT-qPCR | AATCTTTGGAACGGTGGCCT |
| Sequence-based reagent | *mce-1_F* | D2030.5 | RT-qPCR | TGTCCACAAGAACCATGGCT |
| Sequence-based reagent | *mce-1_R* | D2030.5 | RT-qPCR | CGCCGAATGGATGAAGAAGC |

*Appendix 1 Continued on next page*

*Appendix 1 Continued*

| Reagent type (species) or resource | Designation | Source or reference | Identifiers | Additional information |
|---|---|---|---|---|
| Sequence-based reagent | *mmcm-1_F* | ZK1058.1 | RT-qPCR | CAATGTTGCCGATCCTTGGG |
| Sequence-based reagent | *mmcm-1_R* | ZK1058.1 | RT-qPCR | TCCAACAATCACATCTTTTCCAGC |
| Sequence-based reagent | *acdh-1_F* | C55B7.4 | RT-qPCR | TCCGAGCTTCATCCACTTGT |
| Sequence-based reagent | *acdh-1_R* | C55B7.4 | RT-qPCR | CTGACCGAACTGTTCTCTCTGT |
| Sequence-based reagent | *ech-6_F* | T05G5.6 | RT-qPCR | AGGTGGAAACGAGTTGGCAA |
| Sequence-based reagent | *ech-6_R* | T05G5.6 | RT-qPCR | GCTCACAATACCGTGCTCCT |
| Sequence-based reagent | *hach-1_F* | F09F7.4 | RT-qPCR | AGTCATCAGATCGTTCGAGCC |
| Sequence-based reagent | *hach-1_R* | F09F7.4 | RT-qPCR | TCGGTGATTTGTCGGTGAGT |
| Sequence-based reagent | *hphd-1_F* | Y38F1A.6 | RT-qPCR | CAAAGATCTCCACGCCCTGA |
| Sequence-based reagent | *hphd-1_R* | Y38F1A.6 | RT-qPCR | GGAGAGTCCGTGGCAAAGAT |
| Sequence-based reagent | *alh-8_F* | F13D12.4 | RT-qPCR | GGGAGCTCAGGTTCCACTTG |
| Sequence-based reagent | *alh-8_R* | F13D12.4 | RT-qPCR | TGAAGATGGCCGTTCCGTTT |
| Sequence-based reagent | *act-1_F* | T04C12.6 | RT-qPCR | TCGGTATGGGACAGAAGGAC |
| Sequence-based reagent | *act-1_R* | T04C12.6 | RT-qPCR | CATCCCAGTTGGTGACGATA |

