## [Editor Report]

This important work reveals that increased flux towards one carbon metabolism improves neuronal regeneration after injury in *C. elegans*. The presented data are solid and provide compelling support for this conclusion.

---

## [Decision Letter]

**Decision letter after peer review:**

Thank you for submitting your article "O-GlcNAc Signaling Increases Neuron Regeneration Through One-Carbon Metabolism in *Caenorhabditis elegans*" for consideration by *eLife*. Your article has been reviewed by 2 peer reviewers, and the evaluation has been overseen by a Reviewing Editor and Piali Sengupta as the Senior Editor.

Essential revisions:

1. Confirm results with another allele of ogt-1 (possibly a catalytic dead allele as well).

2. Improve the clarity of the presentation: the figures were not always clear, and their schematics of metabolic pathways could certainly be more clear, but also the images of the regenerated axons: it was not clear to me what they are measuring – according to the methods they cut two points and measure regeneration in between, but figures only show one arrow and the axons all look continuous at that point.

3. Either perform experiments OR adapt the data interpretation to disambiguate the following point: The authors use the phrase " enhanced glycolysis" in ogt-1 mutant. They don't show any evidence for this. In the multiply branched pathway(s) of glycolysis the process can be enhanced due to the reduced activity of pyk-1 in got-1 + pyk-1 mutant.

4. There are a number of additional comments and suggestions for improvement listed below.

*Reviewer #2 (Recommendations for the authors):*

This is a potentially important finding regarding the roles of O-GlcNAc cycling and one-carbon metabolism in nerve regeneration. In a previous paper (Taub et al. 2018): they showed: Both ogt-1 and oga-1 mutants show the strong activation of neuronal regeneration phenotype. However, the different biological processes used for the neural regeneration phenotype of between ogt-1 and oga-1 mutant.

The phenotype of ogt-1 mutant for the axon regeneration is that ARK-1 activates AKT-1 to regulate the activation of glycolysis. DAF-16 is not involved in this regeneration process (daf-16 independent). The function of mitochondria (nduf-2.2 and nuo-6 mutant) is assessed in this neural regeneration phenotype of ogt-1. The phenotype of oga-1 mutant for the axon regeneration is dependent upon DAF-16.

A mild modulator of SGK-1 regulated the activity of DAF-16, but not AKT-1. Furthermore, the reduction of mitochondria function (nduf-2.2 mutant) suppressed the neural regeneration of oga-1 mutant. These findings provide the premise for the current work.

My comments about this paper:

Overall, the approach is intriguing and well-documented. The authors find effectors (components of One-carbon metabolism) downstream of the ogt-1 mutant. If they test the following, their current data would be more solid:

The authors use gfat-1 and gfat-2 to mimic the phenotype of ogt-1 mutant, due to the reduction of UDP-GlcNAc for O- GlcNAcylation. However, it is not clear if they tested the phenotype of ogt-1; oga-1 double mutants for neural regeneration. Since they show that both ogt-1 and oga-1 mutants show a similar neuronal regeneration phenotype, but they use different biological pathways. As described above, it is better to assess the neural regeneration phenotype of the oga-1 mutant to determine if high levels of o-glycosylation in the oga-1 mutant occurs. Is the phenotype related to o-glycosylation function in oga-1 mutant? Alternatively, they can check the expression levels of pyk-1 in gfat-1, and gfat-2, and oga-1mutants. The expression level of pyk-1 mutant is down in the ogt-1 mutant by using whole worms, but the authors don't mention the levels of pyk-1 in single cell RNA seq in this paper (Figure 2).

The reviewer checked their single-cell RNA seq data in Supp. Figure 2E and F. None of their selected genes (including pyk-1) which suppress the neural regeneration phenotype of ogt-1 are not listed in this Supp. Figure 2E and F (only pathway analysis by DAVID like enrichment software)? Alternatively, they need to show data for their selected genes.

They also test an atp-3 mutant which affects ATP production. The atp-3 mutant suppressed the neural regeneration phenotype of ogt-1 mutant. This implicates that the overall level of ATP is important to promote the neural regeneration phenotype of ogt-1. (one-carbon metabolism is also linked ATP production). They previously tested nduf-2.2 and nuo-6 mutants for the mitochondrial respiratory chain complex I. However, the lipid consumption in the ogt-1 mutant is greatly impaired, and β-oxidation is important to generate ATP, due to an inefficient glycolysis pathway. They could test the suppression of ogt-1 phenotype by acs-2 and cpt-1 (or other neuronal opt mutant(s)) for the activity of the β-oxidation pathway.

The focus in this paper is on the one-carbon metabolism genes, because their previous paper described that pfk-1.1 and pgk-1 suppress the neural regeneration phenotype of ogt-1. It appears that pyk-1 does not affect ogt-1 phenotype, but pyk-1 by itself shows a similar phenotype to that of ogt-1. According to these results, they focus on the 3-PG branch of the metabolic pathway in the Glycolysis pathway for the neural regeneration phenotype of ogt-1. It would seem important to also see data for the interaction of β-oxidation and ogt-1.

The authors use the phrase " enhanced glycolysis" in ogt-1 mutant. They don't show any evidence for this. In the multiply branched pathway(s) of glycolysis the process can be enhanced due to the reduced activity of pyk-1 in got-1 + pyk-1 mutant. This should be clarified in the discussion of the findings.

Is OGT-1 activity necessary for the phenotype? Were more alleles of ogt-1 tested? OGT alleles both phenotypically and in their extent of penetrance. The authors should at least comment on this in the current paper. Several recent papers have used catalytically dead mutants to assess this. References to these and other relevant papers should be included

In summary, this is a potentially important finding which upon further documentation will be an excellent contribution.

---

## [Author Response]

Essential revisions:1. Confirm results with another allele of ogt-1 (possibly a catalytic dead allele as well).

We have now performed the in vivo regeneration experiment in worms with catalytically dead ogt-1 allele (strain OG1155). Results are similar to those of the *ogt-1* deletion mutation and are now incorporated in the Results section (Figure 1B, line 98-102).

2. Improve the clarity of the presentation: the figures were not always clear, and their schematics of metabolic pathways could certainly be more clear, but also the images of the regenerated axons: it was not clear to me what they are measuring – according to the methods they cut two points and measure regeneration in between, but figures only show one arrow and the axons all look continuous at that point.

Thank you for bringing up this issue. In accordance, we have redrawn the metabolic pathways including all the essential information. We have also included both ablation points in regenerating axon figures and added the details of measurements in the method section (line 523-529).

3. Either perform experiments OR adapt the data interpretation to disambiguate the following point: The authors use the phrase " enhanced glycolysis" in ogt-1 mutant. They don't show any evidence for this. In the multiply branched pathway(s) of glycolysis the process can be enhanced due to the reduced activity of pyk-1 in got-1 + pyk-1 mutant.

Thank you for bringing up this point and we agree that our results do not explicitly demonstrate “enhanced glycolysis”. We have, therefore, removed this phrase throughout the text, referring instead to specific elements of the glycolytic pathway that are implicated. For example, Results section 1 now refers to our specific findings on “Blocking the Hexosamine Biosynthesis Pathway (HBP)”.

4. There are a number of additional comments and suggestions for improvement listed below.

We reviewed all suggestions and comments, have made changes to the manuscript as required and responded to the reviewers below.

Reviewer #2 (Recommendations for the authors):This is a potentially important finding regarding the roles of O-GlcNAc cycling and one-carbon metabolism in nerve regeneration. In a previous paper (Taub et al. 2018): they showed: Both ogt-1 and oga-1 mutants show the strong activation of neuronal regeneration phenotype. However, the different biological processes used for the neural regeneration phenotype of between ogt-1 and oga-1 mutant.The phenotype of ogt-1 mutant for the axon regeneration is that ARK-1 activates AKT-1 to regulate the activation of glycolysis. DAF-16 is not involved in this regeneration process (daf-16 independent). The function of mitochondria (nduf-2.2 and nuo-6 mutant) is assessed in this neural regeneration phenotype of ogt-1. The phenotype of oga-1 mutant for the axon regeneration is dependent upon DAF-16.A mild modulator of SGK-1 regulated the activity of DAF-16, but not AKT-1. Furthermore, the reduction of mitochondria function (nduf-2.2 mutant) suppressed the neural regeneration of oga-1 mutant. These findings provide the premise for the current work.My comments about this paper:Overall, the approach is intriguing and well-documented. The authors find effectors (components of One-carbon metabolism) downstream of the ogt-1 mutant. If they test the following, their current data would be more solid:The authors use gfat-1 and gfat-2 to mimic the phenotype of ogt-1 mutant, due to the reduction of UDP-GlcNAc for O- GlcNAcylation. However, it is not clear if they tested the phenotype of ogt-1; oga-1 double mutants for neural regeneration. Since they show that both ogt-1 and oga-1 mutants show a similar neuronal regeneration phenotype, but they use different biological pathways. As described above, it is better to assess the neural regeneration phenotype of the oga-1 mutant to determine if high levels of o-glycosylation in the oga-1 mutant occurs. Is the phenotype related to o-glycosylation function in oga-1 mutant? Alternatively, they can check the expression levels of pyk-1 in gfat-1, and gfat-2, and oga-1mutants. The expression level of pyk-1 mutant is down in the ogt-1 mutant by using whole worms, but the authors don't mention the levels of pyk-1 in single cell RNA seq in this paper (Figure 2).

We are thankful to the reviewer for such a detailed summarization of our manuscript and insightful suggestions. The reviewer wishes to know the level of O-glycosylation in the *oga-1* mutant. In our previous manuscript (Taub et al. 2018): we have demonstrated that *oga-1* mutation leads to higher levels of O-glycosylation (Taub et al. 2018, Figure 1C). However, our current manuscript is focused on identification of metabolic pathways involved in enhanced neuronal regeneration in *ogt-1* worms. In an effort to maintain the focus of the current study we did not investigate either O-glycosylation levels in the *oga-1,* or expression level of pyk-1/gfat-1/gfat-2 in *oga-1* worms. However, we agree this would be an interesting topic of investigation in future work.

In *ogt-1*, we did observe expression changes in pyk-1 (-1.0 fold), gfat-1 (-1.8 fold) and gfat-2 (-5.4 fold) in neuronal RNAseq data (Table S3, this manuscript). We have now added this information in the manuscript text (Results section 2, line166-167) and comment on their potential relevance.

The reviewer checked their single-cell RNA seq data in Supp. Figure 2E and F. None of their selected genes (including pyk-1) which suppress the neural regeneration phenotype of ogt-1 are not listed in this Supp. Figure 2E and F (only pathway analysis by DAVID like enrichment software)? Alternatively, they need to show data for their selected genes.

We apologize that our presentation of this data was not clear. Figure 2—figure supplement 1E and 1F list only the top 50 up and down regulated genes and therefore do not contain all genes that are differentially regulated. Rather all differentially expressed genes (including pyk-1, gfat-1 and gfat-2) are listed in Table S3 WT-vs-ogt-1 DEGs tab. As mentioned above, we have included these specific results for pyk-1, gfat-1 and gfat-2 in our manuscript text as well. In addition, please note: In this study we have performed neuronal RNAseq but not single cell RNAseq.

They also test an atp-3 mutant which affects ATP production. The atp-3 mutant suppressed the neural regeneration phenotype of ogt-1 mutant. This implicates that the overall level of ATP is important to promote the neural regeneration phenotype of ogt-1. (one-carbon metabolism is also linked ATP production). They previously tested nduf-2.2 and nuo-6 mutants for the mitochondrial respiratory chain complex I. However, the lipid consumption in the ogt-1 mutant is greatly impaired, and β-oxidation is important to generate ATP, due to an inefficient glycolysis pathway. They could test the suppression of ogt-1 phenotype by acs-2 and cpt-1 (or other neuronal opt mutant(s)) for the activity of the β-oxidation pathway.

We thank the reviewer for this insightful comment regarding the β-oxidation pathway. Following this suggestion, we performed regeneration experiments with neuron specific RNAi knockdown against acs-2 and cpt-2 to investigate lipid β-oxidation as a potential link of OCM and ATP generation. However, we did not observe any effect of either acs-2 or cpt-2 knockdown on regeneration in ogt-1 animals (Figure 4—figure supplement 1C). Thus, the β-oxidation pathway does not appear to play an essential role in regeneration which we now present at the end of Results section 4 (line 317-323). Regardless, this additional data adds to the completeness of our study, and we thank the reviewer for their suggestion.

The focus in this paper is on the one-carbon metabolism genes, because their previous paper described that pfk-1.1 and pgk-1 suppress the neural regeneration phenotype of ogt-1. It appears that pyk-1 does not affect ogt-1 phenotype, but pyk-1 by itself shows a similar phenotype to that of ogt-1. According to these results, they focus on the 3-PG branch of the metabolic pathway in the Glycolysis pathway for the neural regeneration phenotype of ogt-1. It would seem important to also see data for the interaction of β-oxidation and ogt-1.

Again, we are thankful for the suggestion regarding the possible link between the β-oxidation pathway and ogt-1. As discussed above, we have now performed regeneration experiments with neuron specific RNAi against acs-2 and cpt-2 to investigate lipid β-oxidation as a potential link of OCM and ATP generation. However, we did not observe any effect on regeneration in ogt-1 animals (Figure 4—figure supplement 1C) suggesting this is not the case.

The authors use the phrase " enhanced glycolysis" in ogt-1 mutant. They don't show any evidence for this. In the multiply branched pathway(s) of glycolysis the process can be enhanced due to the reduced activity of pyk-1 in got-1 + pyk-1 mutant. This should be clarified in the discussion of the findings.

Thank you for bringing this to our attention. We agree that our results do not explicitly demonstrate “enhance glycolysis”. We have changed the phrasing in both the results and Discussion sections to eliminate this claim and explain our findings in a more precise and proper way. We now refer to specific elements of the glycolytic pathway that are directly implicated. Furthermore, we also discuss the plausibility of up regulation of additional glycolytic pathways in ogt-1 mutants and pyk-1 knock down animals in the discussion (line 403-409).

Is OGT-1 activity necessary for the phenotype? Were more alleles of ogt-1 tested? OGT alleles both phenotypically and in their extent of penetrance. The authors should at least comment on this in the current paper. Several recent papers have used catalytically dead mutants to assess this. References to these and other relevant papers should be included

As suggested, we have now performed regeneration experiments with the OGT-1 enzymatic dead allele strain (OG1135) (Figure. 1B, main manuscript Results section 1 line 98-102 and Discussion section line 340-343). Here, we measured similar enhanced regeneration as in ogt-1(deletion) mutant animals, suggesting that it is indeed the loss of OGT-1 enzymatic activity that alters neuron regeneration. These results further support findings in our earlier study, Taub et al. 2018, in which wildtype animals treated with an OGT inhibitor (ST045849; Figure S2A, Taub et al), that reduces enzymatic activity, displayed increased regeneration while similar treatment of ogt-1 animals did not (Figures 1E Taub et al). We further tested the effect of The Hexosamine Biosynthesis Pathway (using RNAi against *gfat-1* and *gfat-2* genes) which is essential to generate UDP-O-GlcNAC, the substrate used by OGT-1 for O-GlcNACylation. This also enhanced regeneration, Figure 1C, suggesting that a shift in metabolic flux towards glycolysis is important. These results are in agreement with our earlier study that ARK-1 and AKT-1 is essential for regeneration in *ogt-1* animals (Taub et al. 2018). The results with the *ogt-1* dead allele further support these conclusions and we are thankful for the suggestion.